# SuperF: Neural Implicit Fields for Multi-Image Super-Resolution

**Sander Riisøen Jyhne**[*]
University of Agder

**Christian Igel**
University of Copenhagen

**Morten Goodwin**
University of Agder

**Per-Arne Andersen**
University of Agder

**Serge Belongie**
University of Copenhagen

**Nico Lang**[*]
University of Copenhagen

## Abstract

High-resolution imagery is often hindered by limitations in sensor technology, atmospheric conditions, and costs. Such challenges occur in satellite remote sensing, but also with handheld cameras like our smartphones. Hence, super-resolution aims to enhance the image resolution algorithmically. Since single-image super-resolution requires solving an inverse problem, these methods must exploit strong priors, e.g. learned from high-resolution training data, or be constrained by auxiliary data, e.g. by a high-resolution guide from another modality. While qualitatively pleasing, such approaches often lead to "hallucinated" structures that do not match reality. In contrast, multi-image super-resolution (MISR) aims to improve the (optical) resolution by constraining the super-resolution process with multiple views taken with sub-pixel shifts. Here, we propose *SuperF*, a test-time optimization approach for MISR that leverages coordinate-based neural networks, also called neural fields. Their ability to represent continuous signals with an implicit neural representation (INR) makes them an ideal fit for the MISR task. The key characteristic of our approach is to share an INR for multiple shifted low-resolution frames and to jointly optimize the frame alignment with the INR. Our approach advances related INR baselines, adopted from burst fusion for layer separation, by directly parameterizing the sub-pixel alignment as optimizable affine transformation parameters and by optimizing via a super-sampled coordinate grid that corresponds to the output resolution. Our experiments yield compelling results on simulated bursts of satellite imagery and ground-level images from handheld cameras, with upsampling factors of up to 8. A key advantage of SuperF is that this approach does not rely on any high-resolution training data.[1]

## 1 Introduction

The spatial resolution of imaging is often limited by sensor capabilities, atmospheric interference, and acquisition costs, affecting various domains including satellite remote sensing, smartphone photography, and medical imaging. Super-resolution (SR) aims to overcome such physical constraints algorithmically. Single-image super-resolution (SISR) methods tackle this inverse problem by relying on strong priors, typically learned from extensive high-resolution (HR) datasets (Ledig et al., 2017; Zhang et al., 2023), or through auxiliary guidance from complementary modalities (De Lutio et al., 2019; 2022; Metzger et al., 2023; Mei et al., 2025). Although SISR methods can produce visually appealing results, their reliance on learned priors often leads to *hallucinated* structures that diverge from the true underlying scene (Cohen et al., 2024). This may be tolerable for smartphone applications, but not for applications in medicine and science.

To mitigate some of these issues, multi-image super-resolution (MISR) has emerged as a special case of super-resolution by incorporating additional information from multiple low-resolution (LR) images captured with slight sub-pixel shifts (Tsai & Huang, 1984; Irani & Peleg, 1991; Elad &

---

[*]Corresponding authors: sander.jyhne@kartverket.no, nila@di.ku.dk
[1]The code, dataset, and demo are available on the project page: sjyhne.github.io/superf

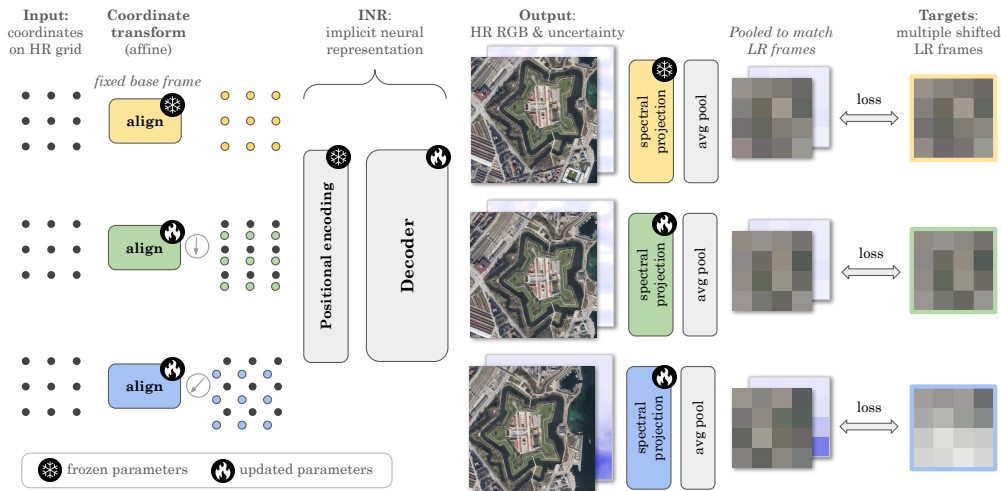

Figure 1: **Illustration of the proposed method.** *SuperF* achieves multi-image super-resolution by sharing an implicit neural representation (INR) across multiple low-resolution (LR) frames with sub-pixel shifts. The LR frames are aligned by jointly optimizing an affine coordinate transformation for each LR frame, together with the parameters of a coordinate-based multi-layer perceptron (MLP) that decodes the input coordinates to RGB values. Hence, leveraging the continuous characteristics of INRs for both the sub-pixel alignment in the pixel coordinate space *and* for representing the underlying high-resolution (HR) signal. For robustness, the proposed INR can optionally represent additional frame-specific uncertainty maps to ignore noisy pixels (e.g. clouds) in the optimization.

Feuer, 1997). As sub-pixel shifts vary across the repeated LR frames, the discretization introduces different aliasing artifacts in each frame. While these artifacts seem to be noise in the LR data, they can be leveraged as complementary information to compute the shared underlying high-resolution image (Wronski et al., 2019).

While MISR can be approached with *supervised learning-based* methods (Bhat et al., 2021a;b) when large training datasets with paired LR and HR data are available, MISR can also be achieved by *test-time optimization (TTO)* approaches that do not require offline training (Wronski et al., 2019; Lafenetre et al., 2023). The latter are particularly interesting, since HR data acquisition is expensive and the creation of large training datasets by pairing of LR images and HR data is non-trivial (Bhat et al., 2021a). Typically, MISR is associated with bursts of images captured in rapid succession. Repeated observations in satellite remote sensing also provide a multi-frame scenario with longer time intervals.[2]

In this work, we introduce *SuperF*, a test-time optimization approach for MISR leveraging the continuous field of implicit neural representations (INR). SuperF shares an INR across multiple shifted LR frames, while jointly estimating the frame-specific alignment. Iteratively refining both the alignment and the shared neural representation effectively reconstructs the underlying high-resolution image on a continuous field (see Fig. 1 for an illustration of the proposed method).

INRs are coordinate-based neural networks, also called neural fields[3], typically parameterized by multi-layer perceptrons (MLPs) that map continuous input coordinates (e.g. 2D image locations) directly to signals like RGB pixel intensities. Optimizing the parameters of such an MLP on an image implicitly encodes the image within its weights. Beyond image representation, INRs have been successfully adopted for data compression (Strümpler et al., 2022; Kwan et al., 2024), 3D shape modeling (Park et al., 2019; Mescheder et al., 2019), novel-view synthesis with neural radiance fields (NeRF) (Mildenhall et al., 2020), and burst fusion for denoising (Pearl et al., 2022) or layer separation of obstructions and background scenes (Nam et al., 2022; Chugunov et al., 2024).

---

[2]Like prior work we use the terms burst and multi-frame interchangeably (Wronski et al., 2019).

[3]We note that the term neural fields is used differently in computational neuroscience (Amari, 1977).

The common unsupervised way to solve the MISR problem is to map the series of LR frames to a HR image, for example using steerable kernel regression (Wronski et al., 2019; Lafenetre et al., 2023). Instead of using the LR frames as an *input* to our model, we draw inspiration from the guided super-resolution work by De Lutio et al. (2019) and turn the problem formulation upside down and treat the LR frames as reconstruction targets. While Nam et al. (2022) have explored such directions for burst fusion and layer separation tasks, their method was not designed to accurately solve sub-pixel frame alignment, which we show is crucial for MISR. Here, we build on these great ideas and design INRs dedicated for the MISR task. By directly parameterizing the affine transformations for the frame alignment and by introducing a supersampling strategy, we improve the sub-pixel alignment and consequently the MISR performance.

We empirically validate the proposed SuperF algorithm on bursts obtained from satellite imagery as well as ground-level images from handheld cameras. In both cases, SuperF gave compelling results. A key aspect of our approach is that it is a TTO method that avoids the need for large amounts of high-resolution training data. This also minimizes the risk of hallucinating high-resolution structures, as opposed to supervised learning based approaches. Our contributions are summarized as:

1. We propose *SuperF*, a test-time optimization method for MISR based on implicit neural representations. We demonstrate that jointly optimizing the sub-pixel frame alignment with an MLP shared across frames is both simple and key to adopt INRs to the MISR task.

2. Our method yields an improved sub-pixel alignment and continuous representation of the high-resolution signal, by directly parameterizing the affine transformations and by optimizing the INR with a supersampling strategy.

3. We introduce SatSynthBurst, a synthetic satellite burst dataset for MISR research and show that our proposed approach generalizes to different domains including ground-level bursts from handheld cameras and satellite image bursts, and is robust to noise in real data.

## 2 RELATED WORK

### 2.1 MULTI-IMAGE SUPER-RESOLUTION (MISR)

Existing MISR approaches contain both learning-free and learning-based methods. They are less prone to hallucinate structures compared to SISR. Test-time optimization (TTO) is applied to exploit natural hand tremors in handheld smartphone photography to capture bursts of slightly shifted raw images. Wronski et al. (2019) propose a steerable kernel regression that enables the direct RGB reconstruction without explicit demosaicing and improving resolution and signal-to-noise ratio, a technology built into the 'pixel' phone. This approach was later reimplemented and adapted for satellite burst applications (Lafenetre et al., 2023).

Like SISR approaches that aim to learn priors from large training datasets, the MISR problem has also been approached with both supervised (Bhat et al., 2021a;b; Cornebise et al., 2022) and self-supervised (Nguyen et al., 2022) learning. Deep neural network architectures for burst super-resolution were proposed to learn the alignment of multiple noisy RAW inputs in latent space via optical flow and the fusion with attention-based modules (Bhat et al., 2021a). Bhat et al. (2021b) proposed a deep reparameterization of the MISR problem and formulated the reconstruction objective in a learned latent space.

In this work, we propose a TTO approach leveraging the continuous nature of INR by jointly optimizing the alignment of low-resolution frames in a continuous coordinate space. Hence, our approach does neither require any high-resolution training data, nor a preprocessing step to register the LR frames. We compare our results with the approach proposed by Lafenetre et al. (2023), which is the closest state-of-the-art TTO approach and, at the same time, the only publicly available implementation. Since their high-resolution test data is not publicly available, we compare results on two different datasets. A handheld burst dataset (Bhat et al., 2021a) and a new synthetic satellite image burst dataset based on open high resolution images (Cornebise et al., 2022). As opposed to prior work that focuses on one domain, we demonstrate that our approach generalizes to different domains including satellite and ground-level image bursts.

## 2.2 Implicit Neural Representations (INR)

Recent advances in INRs have demonstrated the strength of representing continuous signals across various tasks (Essakine et al., 2025), but the development of INRs has mainly been driven by 3D shape modeling (Park et al., 2019; Mescheder et al., 2019) and novel-view synthesis (Mildenhall et al., 2020). These techniques have been adopted for multi-view satellite data (Derksen & Izzo, 2021; Xiangli et al., 2022), species distribution modeling (Cole et al., 2023), and medical imaging to e.g. model 3D MIR volumes (Wu et al., 2021).

Recently INRs have been studied for single-image SR at arbitrary-scales (Chen et al., 2021; Cao et al., 2023; Chen et al., 2023; Zhu et al., 2025). Notably, Chen et al. (2021) propose the Local Implicit Image Function (LIIF) that models RGB values at arbitrary scaling factors. They devised a supervised learning approach that combines the explicit representation from a learned embedding with a local INR that is anchored in the nearest neighbor embeddings. Extending along this line of work, Becker et al. (2025) propose Thera, a neural heat field that explicitly models the point spread function (PSF) to enable analytically correct anti-aliasing at any resolution.

Here we bring forward an approach to leverage the continuous characteristics of INRs for *multi-image* SR. As opposed to prior work on INR for SISR, we do not investigate a supervised approach. Since it is challenging to build training datasets for multi-image super-resolution, we propose a TTO-based solution that does not require any high-resolution training data, but optimizes an INR on multiple shifted LR frames. Key to our proposed approach is the joint optimization of the INR with the alignment of the LR frames, which allows us to share an INR across frames leading to a continuous representation of the underlying high-resolution signal.

Closest to our work is the *NIR* approach presented by Nam et al. (2022). Although originally developed to fuse bursts for layer separation, NIR also serves as an INR baseline for the MISR task. Our method differs in three components. First, while NIR estimates transformation matrices $T_t = g(t)$ using a separate ReLU MLP $g$ conditioned on the frame index $t$, we directly parametrize the transformation matrices as part of the model. Second, to improve sub-pixel alignment, we introduce a supersampling strategy to optimize the model on a high-resolution coordinate grid that is subsequently downsampled for supervision with the LR frames (similar strategies have been proposed to enhance details in novel view synthesis (Wang et al., 2022)). Finally, following prior MISR work (Wronski et al., 2019), rather than estimating transformations for all frames, we use the base frame as the reference coordinate system to relatively align all other LR frames. This reduces the degrees of freedom and facilitates evaluation with high-resolution reference data by avoiding misalignment.

## 3 Methodology

We describe images by functions $[0,1)^d \to \mathbb{R}^{n_c}$ mapping coordinates to intensities. In our application, we consider two-dimensional RGB frames in homogeneous coordinates, i.e., $d = 3$ and $n_c = 3$. Our input are $T$ low-resolution frames $\mathbf{y}_{\text{LR}}^{(1)}, \ldots, \mathbf{y}_{\text{LR}}^{(T)}$ in discretized form, i.e., we are given the values at a finite discrete set of points $\mathcal{W} \subset [0,1)^d$. Our goal is to find an approximation $\hat{\mathbf{y}}_{\text{HR}}$ of the underlying high-resolution signal $\mathbf{y}_{\text{HR}}$ at points $\mathcal{V} \subset [0,1)^d$. Typically, $\mathcal{V}$ and $\mathcal{W}$ are grid points and $|\mathcal{V}| > |\mathcal{W}|$ because the $\mathbf{y}_{\text{LR}}^{(t)}$ are sampled with a lower resolution than the target resolution defined by $\mathcal{V}$.

Our approach is based on the assumption that $\mathbf{y}_{\text{LR}}^{(t)}(\boldsymbol{v}) \approx \varphi * \mathbf{y}_{\text{HR}}(\mathbf{A}^{(t)}\boldsymbol{v})$, where $\mathbf{A}^{(t)}$ is an affine transformation matrix and $\varphi$ is a boxcar filter. Convolution with the boxcar filter implements a spatial average pooling. The affine transformation matrix models misalignments by rotation and translation in the homogeneous coordinate system. In contrast to standard registration methods, our goal is to also exploit misalignments by sub-pixel shifts, i.e., smaller than $\|\mathbf{w}_i - \mathbf{w}_j\|_\infty$ for any $\mathbf{w}_i, \mathbf{w}_j \in \mathcal{W}$.

### 3.1 Implicit neural representation (INR) shared across frames

To optimize an implicit representation of an image, we make use of a coordinate-based multi-layer perceptron (MLP). The MLP model is denoted by $f_{\boldsymbol{\theta}}$ with learnable parameters $\boldsymbol{\theta}$. It is optimized to output the intensities $\hat{\mathbf{y}}$ (e.g., RGB pixel values) for the corresponding input coordinate $\mathbf{v} \in [0,1)^d$.

To share $f_{\boldsymbol{\theta}}$ for $T$ shifted low-resolution frames, we need to *align* them on a sub-pixel scale. To achieve this, we make use of the continuous nature of INRs and optimize the parameters of affine transformation matrices $\hat{\mathbf{A}}^{(t)}$ that are applied to transform the input coordinates for each frame $t$. Following prior work (Wronski et al., 2019), we use the base frame as the reference coordinate system and set $\hat{\mathbf{A}}^{(1)} = I$, where $I$ is the identity matrix (see Fig. 1). The coordinates $\mathbf{v}$ correspond to the high-resolution grid of the base frame:

$$\hat{\mathbf{y}}_{\boldsymbol{\theta}}^{(t)}(\boldsymbol{v}) = \hat{\rho}^{(t)}(f_{\boldsymbol{\theta}}(\hat{\mathbf{A}}^{(t)}\mathbf{v})) \tag{1}$$

The transformation matrices $\hat{\mathbf{A}}^{(t)}$ are directly parameterized by two translation parameters $\Delta x^{(t)}$ and $\Delta y^{(t)}$ as well as one rotation angle $\alpha^{(t)}$ for each frame. In contrast, Nam et al. (2022) proposed to estimate transformation matrices with another MLP for burst fusion. Since we only assume an approximate relationship between LR frames and expect some variation in brightness and contrast, our model optimizes a frame specific spectral projection $\rho^{(t)}$ with a scale and shift parameter per spectral band. For the base frame this projection $\rho^1$ is also fixed (scale 1 and shift 0).

## 3.2 Optimization with low-resolution frames

We propose a *supersampling* strategy to improve the sub-pixel alignment and consequently the implicit neural representation of the HR signal. During optimization, we run the INR at the high-resolution grid $\boldsymbol{v}$ corresponding to the resolution of the super-resolved output. Since we only have the $\mathbf{y}_{\text{LR}}^{(t)}$ available for the optimization, we need to match the output of the INR to the low-resolution frames. That is, we want to find $\boldsymbol{\theta}$ and $\hat{\mathbf{A}}^{(t)}$ such that the low-resolution estimates

$$\hat{\mathbf{y}}_{\text{LR}, \boldsymbol{\theta}}^{(t)}(\boldsymbol{v}) = \varphi * \hat{\rho}^{(t)}(f_{\boldsymbol{\theta}}(\hat{\mathbf{A}}^{(t)}\boldsymbol{v})) \tag{2}$$

equal the low-resolution targets $\mathbf{y}_{\text{LR}}^{(t)}(\boldsymbol{v})$ on $\boldsymbol{v} \in \mathcal{W}$. We fix the boxcar filter $\varphi$, which is implied by different resolutions of the discretized HR output and the given LR images. Ultimately, we optimize for multiple low-resolution frames by averaging a point-wise loss $\ell$ across the $T$ frames:

$$\arg\min \mathcal{L}(\boldsymbol{\theta}, \hat{\mathbf{A}}^{(1)}, \ldots, \hat{\mathbf{A}}^{(T)}) = \frac{1}{T} \sum_{t=1}^{T} \sum_{\boldsymbol{v} \in \mathcal{W}} \ell\left(\hat{\mathbf{y}}_{\text{LR}, \boldsymbol{\theta}}^{(t)}(\boldsymbol{v}), \mathbf{y}_{\text{LR}}^{(t)}(\boldsymbol{v})\right) \tag{3}$$

In practice, the convolution with the boxcar filter and the sampling at grid points $\mathcal{W}$ is simply implemented by an average pooling. We use MLPs with ReLU activation functions and stochastic gradient descent with mini batches of frames.

## 3.3 Uncertainty estimation

To account for uncertainty in the LR frame targets, we model heteroscedastic noise, that is, pixel-wise uncertainty in the LR targets (Nix & Weigend, 1994). In addition to the $n_c$ HR RGB bands, we let the MLP decoder output a band-specific uncertainty estimate per LR frame, i.e. $T \times n_c$ additional outputs. Hence, the total number of outputs returned by the final layer of the MLP equals $(T + 1) \times n_c$. This uncertainty output $\hat{\mathbf{s}}_{\text{LR}}^{(t)}(\boldsymbol{v})$ is interpreted as the logarithm of the variance of a Gaussian random variable centered on the predicted HR signal. Taking the variance estimate into account, optimizing the (logarithmic) likelihood leads to the common loss function:

$$\mathcal{L}_{\text{GNLL}} = \frac{1}{2} \sum_{t=1}^{T} \left[ \hat{\mathbf{s}}_{\text{LR}}^{(t)}(\boldsymbol{v}) + \frac{\left(\hat{\mathbf{y}}_{\text{LR}, \boldsymbol{\theta}}^{(t)}(\boldsymbol{v}) - \mathbf{y}_{\text{LR}}^{(t)}(\boldsymbol{v})\right)^2}{\exp(\hat{\mathbf{s}}_{\text{LR}}^{(t)}(\boldsymbol{v}))} \right], \tag{4}$$

which is referred to as Gaussian negative log-likelihood (GNLL) loss. For locations where the model cannot minimize the squared reconstruction loss, $\mathcal{L}_{\text{GNLL}}$ can further be minimized by increasing the estimated variance. The first term acts as a regularizer and avoids arbitrarily high variance estimates. Both MSE and GNLL are derived under the assumptions that the inputs are independent and that the output has Gaussian noise with constant variance for MSE and varying variance for GNLL, respectively.[4] By switching from MSE to GNLL, we allow the model to minimize the loss by increasing the variance in noisy pixels, instead of forcing it to minimize the reconstruction error.

---

[4]Optimizing MSE is a special case of optimizing GNLL with constant variance.

### 3.4 INPUT TRANSFORMS FOR HIGH-RESOLUTION REPRESENTATIONS

Since we can only optimize the model on low-resolution 'views' of the underlying high-resolution signal, the MLP is prone to output only the low-frequencies of the signal. Hence, to recover high-frequencies captured by the multiple LR frames, we need to steer the MLP to output high-frequency details. In general, coordinate-based MLPs exhibit a spectral bias. The networks prioritize the reconstruction of low-frequency components of the target signal, whereas high-frequency details emerge only slowly during the convergence of optimization. Several approaches have been proposed to overcome this spectral bias (Sitzmann et al., 2020; Saragadam et al., 2023). We rely on the commonly used Fourier features (Tancik et al., 2020) as a positional encoding. The feature map $\gamma : [0, 1)^d \to \mathbb{R}^{2m}$ is based on a random set of sine and cosine basis functions:

$$\gamma(\mathbf{v}) = \left[ \cos(2\pi \mathbf{b}_1^{\mathrm{T}} \mathbf{v}), \ldots, \cos(2\pi \mathbf{b}_m^{\mathrm{T}} \mathbf{v}), \sin(2\pi \mathbf{b}_1^{\mathrm{T}} \mathbf{v}), \ldots, \sin(2\pi \mathbf{b}_m^{\mathrm{T}} \mathbf{v}) \right]^{\mathrm{T}} \tag{5}$$

Each $\mathbf{b}_i \in \mathbb{R}^d$, $i = 1, \ldots, m$, is sampled from an isotropic multivariate Gaussian distribution $\mathcal{N}(0, \sigma^2 \mathbf{I})$, where the scale $\sigma$ is a hyperparameter controlling the range of the sampled frequencies. We show that this hyperparameter is sensitive to the domain (e.g., satellite images vs. ground-level burst images), but the same value performs well across all samples within a domain.[5]

## 4 EXPERIMENTAL RESULTS AND DISCUSSION

### 4.1 DATASETS

Our experiments are based on datasets from two domains: remote sensing and handheld cameras (see examples in Fig. 2). First, we create a synthetic burst dataset from high-resolution satellite images to study various characteristics and the sensitivity of our proposed approach. Second, we demonstrate that our SuperF approach also generalizes to ground-level bursts from handheld cameras.

**SatSynthBurst (satellite imagery).** To study MISR for satellite imagery bursts we constructed a synthetic burst dataset derived from 20 open high-resolution satellite images selected from the WorldStrat dataset (Cornebise et al., 2022) (see examples in Appendix). For each high-resolution sample, we generate 16 low-resolution frames with scale factors 2, 4, and 8, by randomly sampling sub-pixel shifts. Our dataset provides one LR *base frame* that is spatially aligned with the HR test images, a missing feature of existing datasets. This framework allows to study the influence of different upsampling factors keeping the HR resolution fixed. Furthermore, it gives us control over the sub-pixel shifts, and noise intensity. Although our approach does not require the true misalignment parameters, they allow us to monitor the optimization dynamics when estimating the alignment. To realistically simulate the image formation process, we use spectral variations and additive Gaussian noise in all experiments (if not further specified). We follow the best practices for generating synthetic super-resolution data to mimic the modulation transfer function (mtf) of the Sentinel-2 sensor described by Lanaras et al. (2018). Details in the Appendix section A.1.

**SyntheticBurst (ground-level imagery).** To evaluate on handheld bursts of ground-level scenes, we make use of the SyntheticBurst data provided by Bhat et al. (2021a) and e.g. used by Bhat et al. (2021b). We select 50 out of the 300 provided ground level bursts that provide interesting high-resolution structures, i.e., we remove for example bursts that are crops of homogeneous areas such as building walls or skies. Each burst consists of 14 LR frames, originally at a scale factor of $\times 8$. To study different upsampling factors, we vary the HR output resolution by downsampling the HR reference images, but we keep the LR frames as provided to avoid changing the underlying noise model. Since this dataset does not provide a base LR frame aligned with the HR test image, we run a brute force postprocessing to improve the alignment of the predictions before computing the error metrics (see Appendix section A.3).

---

[5]Tancik et al. (2020) establish the relation between $\sigma^2$ and the bandwidth of the neural tangent kernel modeling the resulting MLP. They argue that a wider kernel supports the learning of high frequency components, but that a too wide kernel can lead to aliasing artifacts. They conclude that the parameter is problem dependent and has to be tuned.

## 4.2 EXPERIMENTAL SETUP

We follow standard practices in super-resolution and report Peak Signal-to-Noise Ratio (PSNR), Structural Similarity Index Measure (SSIM), and Learned Perceptual Image Patch Similarity (LPIPS) (using AlexNet), computed using the implementation by Bhat et al. (2021a). All experiments are implemented in PyTorch and executed on a single NVIDIA H100 GPU with 80 GB of VRAM (see table 8 in the Appendix for detailed training time, speed, memory usage, and FLOPs). If not further specified, all experiments use the AdamW optimizer with a base learning rate of $2 \times 10^{-3}$, which is decayed to $1 \times 10^{-6}$ over 2000 iterations using a cosine annealing schedule and a batch size of 1 frame.

During evaluation, a 16-pixel boundary is cropped from all sides to reduce edge artifacts. We additionally apply color matching as a post-processing step, following Bhat et al. (2021a), to correct for global color and intensity shifts between the reconstruction and the ground truth. The scale hyperparameter of the Fourier feature positional encoding is set to 10 for the SatSynthBurst and to 3 for the SyntheticBurst dataset.

## 4.3 COMPARISON TO EXISTING TEST-TIME OPTIMIZATION APPROACHES

**Baseline approaches.** As we propose a TTO approach, we compare to a state-of-the-art TTO approach for MISR, a steerable kernel regression method by Lafenetre et al. (2023), which is an adapted version of the approach described by Wronski et al. (2019). We also compare to a burst fusion approach by Nam et al. (2022) (named NIR) and adapt it as a MISR baseline. Although developed for burst fusion for layer separation tasks, it is related to our method as it uses an INR with a built-in frame alignment (as introduced in section 2.2). To study the effect of each proposed methodological component, we integrate the NIR approach in our framework to keep all other components, that are design choices, the same. Thus, for both our SuperF and NIR, we use i) the same INR encoder (i.e., Fourier features with a ReLU MLP instead of Siren), ii) an affine matrix (instead of a homography), iii) the same batch optimization, and iv) the same frame-specific spectral projection. As proposed by Nam et al. (2022), we run NIR for up to 5k iterations. For reference, we also report the performance of a bilinear upsampling of the LR base frame.

**Comparison results.** We present quantitative results in Table 1 for upsampling factors $\times 2$, $\times 4$, $\times 8$ focusing on PSNR. Additional metrics including LPIPS and SSIM are provided in the Appendix Table 5. We find that both existing approaches cannot outperform the bilinear baseline in terms of PSNR. However, Lafenetre et al. (2023) yields better LPIPS than bilinear upsampling for an upsampling factor $\times 4$. While Lafenetre et al. (2023) has been designed for MISR of satellite bursts for upsampling up to factor 2, the NIR approach by Nam et al. (2022) has not been explicitly designed for super-resolution, but for denoising and layer-separation at the original resolution of the LR frames. In contrast, our SuperF approach outperforms all baselines using both the MSE and GNLL loss. While on SyntheticBurst both losses perform on par, the GNLL loss consistently improves upon the MSE on SatSynthBurst. The GNLL possibly allows the INR to be more robust against the spectral variation in this dataset, which is less pronounced in the ground-level bursts.

Qualitatively, the methods by both Lafenetre et al. (2023) and Nam et al. (2022) seem to be able to smooth and hence denoise the ground-level bursts, but lead to overly smooth results (see Fig. 2 and Appendix Fig. 11–13 for upsampling factors $\times 2$, $\times 4$, $\times 8$). Furthermore, we observed that for some satellite scenes, NIR produces a constant output and collapses at the beginning of the optimization. Our proposed SuperF approach yields pleasing results that can deal with the high noise-level in the ground-level bursts and represent the high-resolution signal in satellite scenes.

## 4.4 ABLATION STUDIES AND SENSITIVITY ANALYSES

To understand the individual components of our proposed methodology, we evaluate the performance gradually turning on each component in Table 2. We start with using our implementation of the INR with just a single LR frame (first row), but compare the resulting high-resolution reconstruction. This base experiment corresponds to an INR *without* using i) the Fourier feature positional encoding (FF); ii) the multiple LR frames (multi-frame); iii) the optimization of the alignment of the LR frames (align);

Table 1: **Comparison with TTO baselines.** PSNR (↑) for different upscaling factors. Note that the SatSynthBurst fixes the HR output resolution while SyntheticBurst fixes the resolution of the LR frames. Hence, as the upsampling factor increases, we only expect lower performance metrics for SatSynthBurst. *Experimental setup*: upsampling factors ×2, ×4, ×8, 16 LR frames. Standard deviation across samples is given in parentheses and the number of iterations in square brackets.

| | SatSynthBurst | | | SyntheticBurst | | |
|---|---|---|---|---|---|---|
| | ×2 | ×4 | ×8 | ×2 | ×4 | ×8 |
| Bilinear | 34.69 (3.50) | 29.71 (3.64) | 26.62 (3.68) | 27.66 (3.50) | 26.12 (3.72) | 25.44 (3.82) |
| Lafenetre et al. (2023) | 33.46 (3.62) | 27.70 (3.79) | 24.88 (3.71) | 27.02 (3.29) | 26.46 (3.05) | 25.19 (2.97) |
| NIR (Nam et al., 2022) [2k] | 26.26 (3.91) | 24.63 (4.41) | 23.85 (3.79) | 23.62 (4.43) | 22.69 (4.41) | 22.28 (4.40) |
| NIR (Nam et al., 2022) [5k] | 25.65 (5.82) | 24.99 (4.12) | 23.61 (2.97) | 24.46 (4.31) | 23.39 (4.32) | 22.93 (4.33) |
| SuperF MSE (ours) [2k] | 36.73 (1.66) | 32.94 (1.83) | 28.87 (2.32) | 29.38 (3.43) | 27.90 (3.94) | 27.08 (3.97) |
| SuperF GNLL (ours) [2k] | 37.26 (2.30) | 34.03 (2.71) | 29.28 (3.21) | 29.48 (3.76) | 27.47 (4.18) | 26.58 (4.18) |

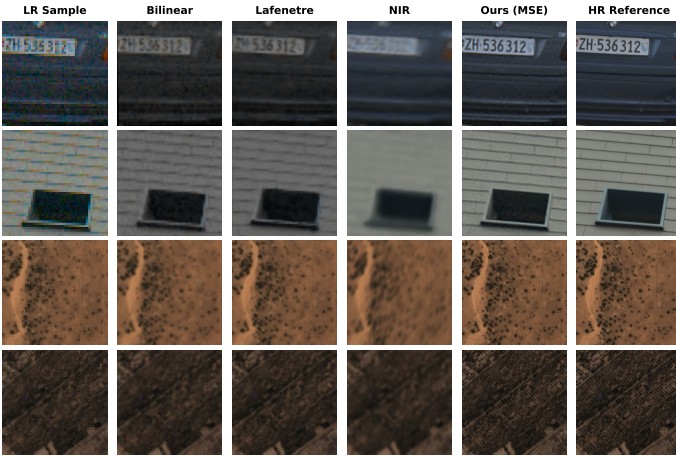

Figure 2: **Qualitative comparison with upsampling factor ×4.** From left to right, we show: one low-resolution (LR) frame, bilinear upsampling, steerable kernel regression (Lafenetre et al., 2023), NIR (Nam et al., 2022), our *SuperF* approach, and the high-resolution (HR) reference. Samples are from SyntheticBurst (row 1, 2) and SatSynthBurst (row 3, 4). More results in Appendix Fig. 11–13.

We confirm that the positional encoding is also a crucial aspect for INRs in the MISR setup. However, optimization on a single LR frame is not able to recover any high-resolution details and, in fact, performs similar to a bilinear upsampling when used with the FF encoding (comparing Table 2 with Appendix Table 5). Next, we evaluate the performance of optimizing a shared INR on multiple LR frames without optimizing their alignment (third row). Compared to optimizing with a single LR frame, this leads to an even lower performance – a result of the sub-pixel shifts which blurs the signal. Only by optimizing the sub-pixel alignment together with the shared INR, our proposed approach is able to leverage the information in multiple LR frames, which leads to substantial improvement in performance (fourth row). These results hold systematically for both domains, satellite images (SatSynthBurst) and ground-level bursts (SyntheticBurst).

**Effect of the proposed components.** To understand which of the proposed components lead to an advancement over the closest method by Nam et al. (2022), we investigate the effect by turning each component on and off individually in Table 3. We find that using a direct parametrization of the affine transformation parameters, instead of using another MLP to estimate the transformation is most crucial, followed by the supersampling strategy, to reduce the sub-pixel alignment error and MISR performance. Fixing the base frame is also beneficial, but only when using our direct parametrization. Combined, all three components substantially improve super-resolution performance and reduce the sub-pixel alignment error (euclidean distance).

**Sensitivity analyses** Our method mainly depends on one key hyperparameter, the scale of the Fourier features $\sigma^2$, as described in section 3.4. We show that our algorithm is sensitive to this

Table 2: **Ablation studies.** We study the importance of the individual components of our proposed approach. The base experiment (first row) corresponds to an INR *without*: i) the Fourier feature positional encoding (FF); ii) using multiple LR frames, i.e. a single frame (multi-frame); iii) optimizing the alignment of the LR frames (align); *Experimental setup*: upsampling factor $\times 4$, 16 LR frames. Standard deviation across samples shown in parentheses.

| FF | multi-frame | align | SatSynthBurst | | | SyntheticBurst | | |
|---|---|---|---|---|---|---|---|---|
| | | | PSNR ↑ | SSIM ↑ | LPIPS ↓ | PSNR ↑ | SSIM ↑ | LPIPS ↓ |
| ✗ | ✗ | ✗ | 20.33 (2.16) | 0.337 (0.160) | 0.661 (0.050) | 16.63 (2.91) | 0.225 (0.138) | 0.720 (0.042) |
| ✓ | ✗ | ✗ | 30.42 (3.24) | 0.774 (0.073) | 0.361 (0.047) | 24.69 (3.08) | 0.546 (0.107) | 0.573 (0.047) |
| ✓ | ✓ | ✗ | 28.11 (3.32) | 0.663 (0.120) | 0.458 (0.051) | 22.83 (3.81) | 0.523 (0.153) | 0.592 (0.050) |
| ✓ | ✓ | ✓ | 32.94 (1.83) | 0.853 (0.025) | 0.287 (0.035) | 27.87 (3.92) | 0.774 (0.102) | 0.383 (0.070) |

Table 3: **Effect of the proposed components** to advance the NIR baseline (Nam et al., 2022) on the SatSynthBurst dataset. The first row is the NIR baseline without any of our contributions. Variants incrementally add or remove: i) direct parameterization of $T$ ("Direct $T$"), ii) optimizing with supersampling ("SS"), and iii) using a fixed base frame ("FBF").

| Method | Direct $T$ | SS | FBF | PSNR ↑ | SSIM ↑ | LPIPS ↓ | Align. Err. ↓ | Iter. |
|---|---|---|---|---|---|---|---|---|
| NIR (Nam et al., 2022) | ✗ | ✗ | ✗ | 24.63 | 0.539 | 0.595 | 0.650 | 2000 |
| | ✓ | ✗ | ✗ | 26.14 | 0.580 | 0.479 | 0.012 | 2000 |
| | ✗ | ✓ | ✗ | 24.76 | 0.482 | 0.593 | 0.079 | 2000 |
| | ✗ | ✗ | ✓ | 26.39 | 0.621 | 0.483 | 0.319 | 2000 |
| | ✗ | ✓ | ✓ | 24.76 | 0.482 | 0.593 | 1.324 | 2000 |
| | ✓ | ✗ | ✓ | 26.20 | 0.578 | 0.476 | 0.012 | 2000 |
| | ✓ | ✓ | ✗ | 31.30 | 0.818 | 0.295 | 0.012 | 2000 |
| SuperF (ours) | ✓ | ✓ | ✓ | 32.94 | 0.853 | 0.287 | 0.012 | 2000 |

parameter in Appendix Fig. 7 and 8 and that the best Fourier feature scale depends on the domain. For the satellite image bursts an optimal scale is 10 and for the ground-level bursts it is 3. However, the optimal setting does not depend on the loss and the same setting generalizes across samples in the same domain. Furthermore, the sensitivity to the number of LR frames is shown in the Appendix section C.2.

## 4.5 RESULTS ON REAL SENTINEL-2 SATELLITE IMAGES

We demonstrate that our method can be applied to real-world satellite images from Sentinel-2 (see Fig. 3). We use Sentinel-2 images available from an AWS STAC endpoint[6], and use the cloud-free samples for super-resolution. These are real-world examples and are therefore affected by noise due to lighting variation, changing landcover (e.g. crops), or seasonal variations like snow cover. In scenarios where the noise is dominating, our assumption of repeated observations of the same scene does not hold and further development is needed to account for such high noise levels. [7]

**Robustness to occlusions.** To study robustness of our approach for noisy satellite image time series where some LR frames are partially covered by clouds, we curated two subsets of the WorldStrat dataset (Cornebise et al., 2022) using real Sentinel-2 images, *WorldStrat-sweet* and *WorldStrat-bitter*, with clean and noisy time series, respectively. On the *bitter* dataset, we find that the GNLL approach effectively ignores cloudy pixels (Appendix Fig. 10) and improves performance compared to the MSE loss (Appendix Table 7). For clean time series in *sweet*, both losses perform on par.

## 4.6 DISCUSSION

Our proposed MISR approach, which jointly optimizes the alignment of LR frames and a shared INR, exhibits several advantageous characteristics: i) While existing approaches require a pre-

---

[6]github.com/Element84/earth-search (accessed: 2025-11-01)

[7]A demo app to super-resolve any place on Earth is available here: sjyhne.github.io/superf.

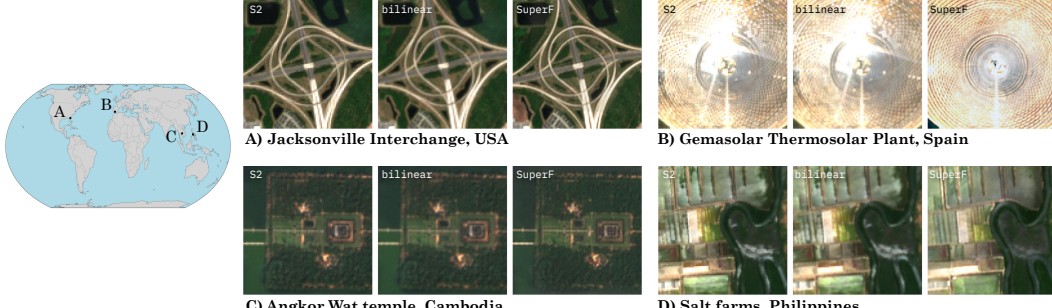

Figure 3: **Qualitative examples using real satellite images.** We demonstrate that our method can align and super-resolve real satellite images from the Sentinel-2 mission by an upsampling factor of 5 using a filtered time series from Sentinel-2. Depending on the cloud cover this leads to a varying number of LR images retrieved within 3–5 months (number of images: A: 25, B:15, C:9, D:7).

alignment step, SuperF directly works on large shifts by optimizing the alignment in continuous coordinate space (see Appendix section C.4). ii) As a TTO approach, there is no need for any high-resolution training data. This allows SuperF to be applied to new domains without any pretraining. However, some limitations exist.

**Limitations.** *Runtime* may limit certain applications. Although our compact MLP is fairly memory-efficient, the iterative optimization process takes several seconds in our experiments (see Appendix Table 8, running non-optimized code). This may pose limitations for mobile device applications, but is less critical for remote sensing scenarios and other scientific and medical applications. A possible way to reduce the number of iterations needed may be to learn the initialization of the INR as shown by Tancik et al. (2021). *Real-world data* can be highly noisy. For instance, satellite imagery may also partially be affected by cloud cover and handheld ground-level bursts may depict changing scenes. We assume that the observations capture the same scene. Occlusions and other drastic changes between frames introduce noise, which requires further analyses. However, our uncertainty estimation module may help to be robust to such noise (Appendix Fig. 10). *Risk of overfitting* increases when setting the Fourier features scale hyperparameter too high. While this parameter depends on the domain, it is rather robust across samples within a domain (Appendix Fig. 7). In practice, this parameter can be tuned by using some high-resolution validation images as shown in Fig. 7 or via visual inspection (see Fig. 8). Our proposed methodology would allow to test alternative INR decoders that avoid positional encodings such as SIREN (Sitzmann et al., 2020) or WIRE (Saragadam et al., 2023).

**Impact on society.** The ability to super-resolve publicly available satellite data like Sentinel-2 enables a vast range of applications anywhere on Earth. This approach can support efforts to address critical societal challenges such as climate adaptation, biodiversity conservation, and food security, for example, by facilitating environmental monitoring of deforestation, tree cover, tree counting, and mapping agricultural fields. However, these technological advances also carry the potential for misuse, including in the context of geopolitical conflicts or resource exploitation.

## 5 CONCLUSION

We bring forward an approach to leverage the continuous characteristics of implicit neural representations for multi-image super-resolution. The key characteristic of *SuperF* is to jointly optimize the sub-pixel alignment of multiple low-resolution frames while sharing an INR across all frames. SuperF improves upon existing INR-based burst fusion approaches by optimizing INRs with a direct parameterization of the affine transformations and using a supersampling strategy, which leads to improved sub-pixel alignment and thus MISR performance. As a TTO method, SuperF does not require any high-resolution training data, which facilitates the applicability to new domains — e.g. any location on Earth — and minimizes the risk of hallucinating high-resolution structures.

## ACKNOWLEDGMENTS

This work was supported in part by the Pioneer Centre for AI, DNRF grant number P1 and by the Global Wetland Center (grant number NNF23OC0081089) from Novo Nordisk Foundation.

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

# A  DATASET CREATION AND EVALUATION PROCEDURE

In this section we provide details on i) the downsampling of high-resolution satellite images to create synthetic bursts of slightly shifted low-resolution images and ii) the postprocessing needed for evaluating the predicted high-resolution images.

## A.1  CREATION OF THE SATSYNTHBURST DATASET (SATELLITE IMAGERY)

We constructed a synthetic burst dataset derived from 20 open high-resolution satellite images selected from the WorldStrat dataset (Cornebise et al., 2022) (see examples in Fig. 4). The high-resolution images from Airbus SPOT 6/7 satellite with a ground sampling distance (GSD) of up to 1.5 m are published under a CC BY-NC 4.0 license[8], which allows us to publicly redistribute our SatSynthBurst datasets under the same license for non-commercial purposes. We aim to simulate low-resolution images comparable to the Sentinel-2 mission, but at varying spatial resolutions allowing to study downsampling factors $s$ of 2, 4, and 8. To simulate variation in the imaging conditions that could occur between images captured over several weeks, we incorporate spectral augmentations and additive Gaussian noise. Additionally, we follow the work by Lanaras et al. (2018) for generating synthetic super-resolution data using the modulation transfer function (*mtf*) of the Sentinel-2 sensor. Hence, before downsampling, we blur the high-resolution images with a Gaussian filter of standard deviation $u = 1/s$ pixels, which emulates the *mtf* of Sentinel-2 and, thus, the effective point spread function (*psf*) which is described as $psf = \sqrt{-2\log(mtf)/\pi^2}$. This is followed by an average pooling with a window size of $s \times s$.

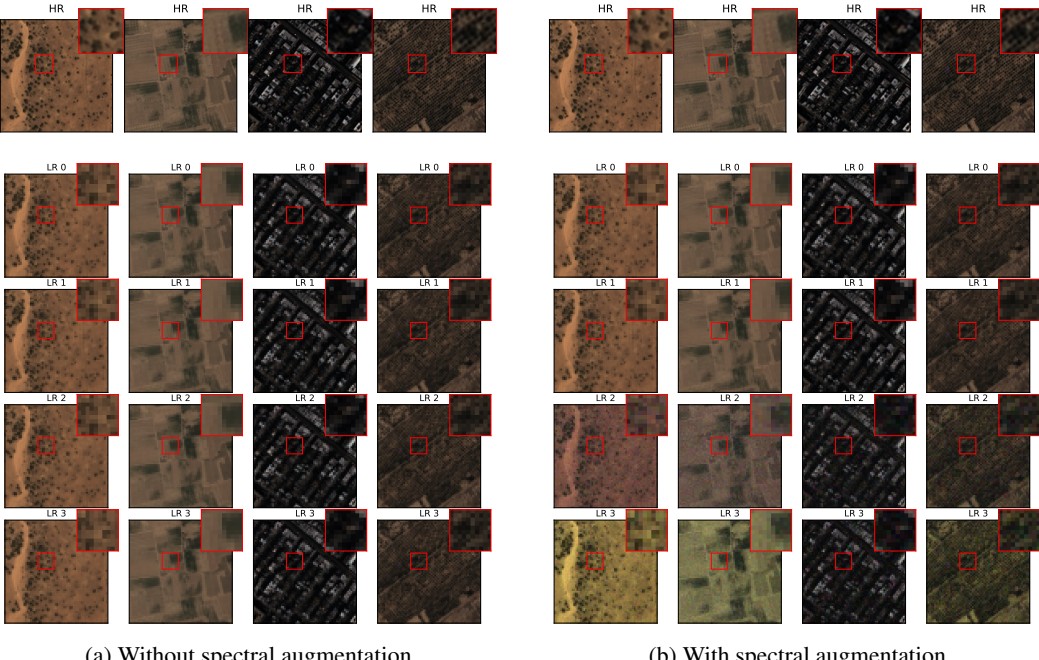

(a) Without spectral augmentation          (b) With spectral augmentation

Figure 4: **Examples of the SatSynthBurst dataset (factor ×4).** The top row shows the underlying high-resolution (HR) image. Below we show four slightly misaligned low-resolution (LR) frames.

## A.2  POSTPROCESSING FOR EVALUATION

We follow common practice in evaluating MISR results and use a spectral alignment proposed by Bhat et al. (2021a) to correct any spectral mismatch between the high-resolution prediction and the test image. Metrics like the PSNR and SSIM are rather sensitive to small misalignments, whereas LPIPS is more robust. Furthermore, we follow the evaluation protocol of Bhat et al. (2021a) and

---

[8]https://creativecommons.org/licenses/by-nc/4.0/ (accessed: 2025-05-20)

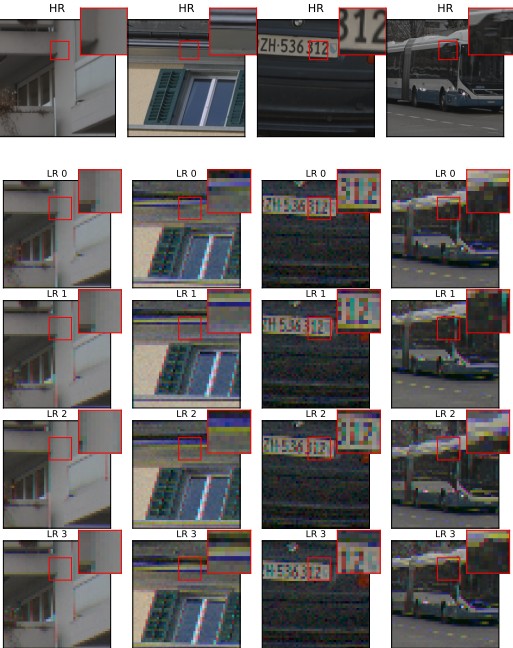

Figure 5: **Examples of the SyntheticBurst dataset (factor ×8).** The top row shows the underlying high-resolution (HR) image. Below we show four slightly misaligned low-resolution (LR) frames.

mask out a buffer of 16 boundary pixels to avoid the effect of any boundary artifacts in the dataset (specifically, the SyntheticBurst dataset).

### A.3 EVALUATION ON SYNTHETICBURST (GROUND-LEVEL IMAGERY)

Unlike our SatSynthBurst dataset, the SyntheticBurst dataset (Bhat et al., 2021a) does not provide a base frame which is spatially aligned with the high-resolution test image (see examples in Fig. 5). Thus, an additional postprocessing step is needed, before a predicted HR image can be evaluated on the given HR test image. Therefore, we employ a brute force spatial alignment strategy to align the predicted image with the test image using an affine transformation consisting of rotation and a spatial translation. Our strategy selects the optimal translation within a 4×4 pixel neighborhood and the optimal rotation angle within a range of $[0, 4]$ degrees.

We have experimented with the alignment strategy presented by Bhat et al. (2021a), that uses a trained PWC-Net (Sun et al., 2018) to estimate the optical flow from the prediction to the reference test image. However, this led to artifacts in the warped prediction, which is why we chose the brute force postprocessing.

### B IMPLEMENTATION DETAILS

We summarize the hyperparameter settings for both datasets in Table 4. The only hyperparameter that differs between datasets is the Fourier feature scale.

### C ADDITIONAL RESULTS

#### C.1 COMPARISON OF BASELINES WITH ADDITIONAL METRICS

We provide additional evaluation metrics including PSNR, SSIM, and LPIPS for the baseline comparison in Table 5.

Table 4: **Hyperparameter settings.**

| Hyperparameters | SatSynthBurst | SyntheticBurst |
|---|---|---|
| LR resolution | 128 / 64 / 32 | 48 |
| HR resolution | 256 | 96 / 192 / 384 |
| Optimizer | AdamW | |
| Learning rate sched. | Cosine annealing | |
| Learning rate base | $2 \times 10^{-3}$ | |
| Learning rate min | $1 \times 10^{-6}$ | |
| Weight decay | 0.05 | |
| Adam $\beta$ | (0.9, 0.999) | |
| Batch size | 1 LR frame per iteration | |
| Training iterations | 2000 | |
| Fourier feature scale ($\sigma$) | 10 | 3 |
| positional encoding dim | 256 | |
| INR decoder | MLP (4 layers, ReLU, dim=256) | |

Table 5: **Comparison of test-time optimization methods.** Ours uses Fourier feature with scale 10 for SatSynthBurst (satellite) and scale 3 for SyntheticBurst (ground-level). *Experimental setup*: upsampling factor $\times 4$, 16 LR frames. Standard deviation across samples is given in parentheses and the number of iterations in square brackets.

| | SatSynthBurst | | | SyntheticBurst | | |
|---|---|---|---|---|---|---|
| | PSNR ↑ | SSIM ↑ | LPIPS ↓ | PSNR ↑ | SSIM ↑ | LPIPS ↓ |
| Bilinear | 29.71 (3.64) | 0.746 (0.104) | 0.382 (0.043) | 26.12 (3.72) | 0.703 (0.121) | 0.455 (0.077) |
| Lafenetre et al. (2023) | 27.70 (3.79) | 0.680 (0.130) | 0.261 (0.055) | 26.46 (3.05) | 0.664 (0.121) | 0.384 (0.118) |
| NIR (Nam et al., 2022) [2k] | 24.63 (4.42) | 0.539 (0.175) | 0.595 (0.076) | 22.69 (4.41) | 0.576 (0.171) | 0.616 (0.089) |
| NIR (Nam et al., 2022) [5k] | 24.99 (4.13) | 0.544 (0.167) | 0.587 (0.082) | 23.39 (4.32) | 0.606 (0.165) | 0.574 (0.090) |
| SuperF MSE (ours) [2k] | 32.94 (1.83) | 0.853 (0.035) | 0.287 (0.054) | 27.90 (3.95) | 0.774 (0.102) | 0.385 (0.070) |

## C.2 SENSITIVITY TO THE NUMBER OF LR FRAMES

We study the sensitivity to the number of available LR frames in Fig. 6. This is a critical aspect for both application domains. Handheld bursts might be limited in the number of frames since the scene might change for long overall exposure times. Satellite imagery like Sentinel-2 are captured with a revisit period of $\approx 5$ days. We thus need to consider longer time windows to obtain multi-frame satellite images. However, longer time windows may lead to changing appearance of the scene due to activity on the ground or seasonality, which will hinder MISR. Furthermore, cloud-free images may be scarce, depending on the geographic region.

For the SatSynthBurst dataset, we observe that the PSNR saturates with 8 samples for the factor $\times 2$, but keeps increasing slightly when using 16 samples for the larger upsampling factors. In contrast, the PSNR for the ground-level bursts keeps improving with more frames even for the factor $\times 2$. Additional frames may help to reduce the high noise level in the ground-level bursts.

## C.3 SENSITIVITY TO THE FOURIER FEATURE SCALE PARAMETER

As shown in the main paper, our method is sensitive to the Fourier feature scale, and the optimal hyperparameter depends on the domain, i.e. satellite imagery and ground-level bursts. We show the qualitative effect of the different Fourier features scales in Fig. 8. Setting the scale too low leads to over-smoothing, whereas setting it too high leads to grainy artifacts. However, we find that a single parameter setting performs well across samples within a domain. We use the optimal setting for upsampling factor 4 for all experiments including factor 2 and 8 (see hyperparameter setting in Table 4).

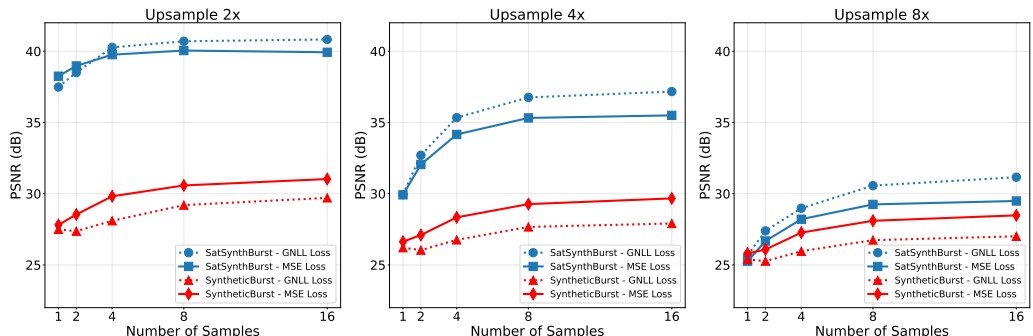

Figure 6: **Sensitivity to the number of LR frames.** From left to right, we report PSNR for upsampling factors 2, 4, and 8 by varying the number of LR frames on the horizontal axis.

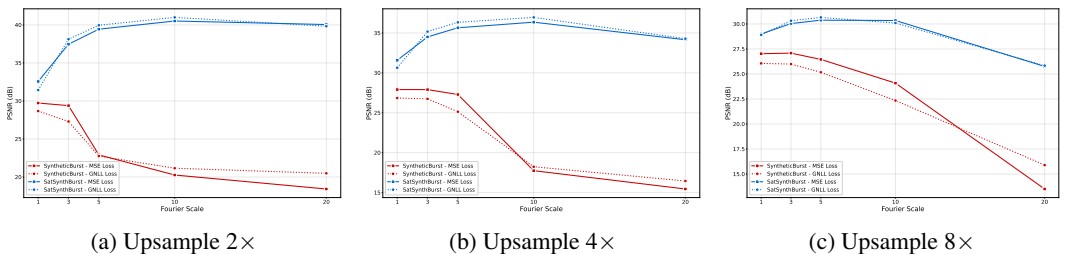

(a) Upsample 2×                    (b) Upsample 4×                    (c) Upsample 8×

Figure 7: **Sensitivity analysis of the Fourier feature scale.** The optimal hyperparameter depends on the domain, i.e. satellite imagery (SatSynthBurst in blue) and ground-level bursts (SyntheticBurst in red) require different settings. However, the optimal setting is invariant to the loss. For SyntheticBurst we see a small difference between the upsampling factor experiments. However, we note that the two datasets differ in the strategy of creating different upsampling factors. The SyntheticBurst varies the HR output resolution with a fixed resolution of the LR frames. Hence, the absolute output resolution may affect the optimal Fourier feature scale. In contrast, the SatSynthBurst dataset fixes the HR output resolution and varies the resolution of the LR frames.

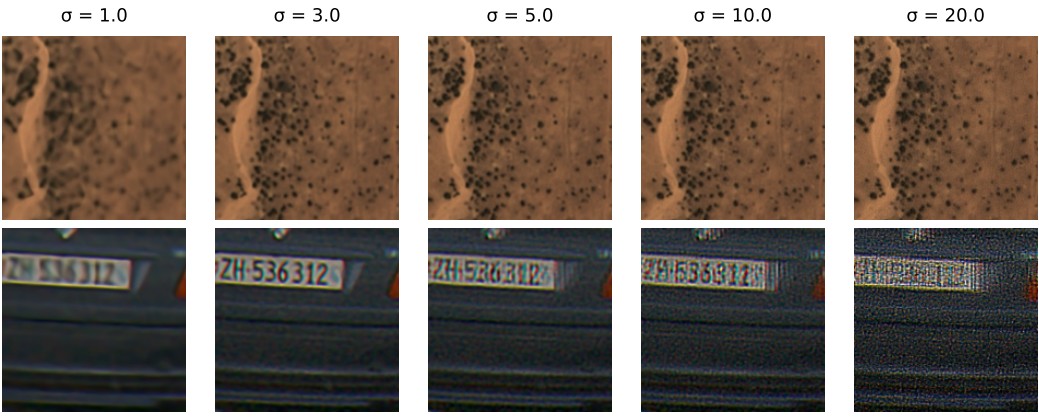

Figure 8: **Effect of the Fourier feature scale $\sigma$ for MSE (upsampling factor $\times 4$).** The optimal hyperparameter depends on the domain, i.e. satellite imagery and ground-level bursts require different settings. Setting the scale too low leads to over-smoothing, whereas setting it too high leads to grainy artifacts.

## C.4 ALIGNING LARGE SHIFTS WITHOUT PREPROCESSING

While existing methods (Wronski et al., 2019; Lafenetre et al., 2023) rely on a pre-alignment procedure to first reduce the misalignment to sub-pixel shifts, our method can directly work on multi-pixel shifts. Although the PSNR drops consistently with increased shifts, when comparing the results of bursts with sub-pixel shifts against bursts with shifts up to 4 LR pixels (see Table 6), our method keeps outperforming the baseline. However, our method breaks in the extreme case with upsampling factor 8, i.e. the shifts of 4 LR pixels correspond to 32 pixels in the high-resolution image of size 256×256 pixels (i.e., >12% relative shift). Further investigation is needed to study if this issue could be resolved with a different hyperparameter setting, e.g. by increasing the number of iterations or the learning rate.

Table 6: **Comparing sub-pixel with large misalignments (PSNR).** We compare our SuperF results on bursts with sub-pixel shifts, i.e. max shifts of 1.0 LR pixels, vs. bursts with large shifts up to 4.0 LR pixels. *Experimental setup*: upsampling factor ×2, ×4, ×8; 16 LR frames. Standard deviation across samples shown in parentheses.

|  |  | SatSynthBurst | | |
|  |  | ×2 | ×4 | ×8 |
|---|---|---|---|---|
|  | Bilinear | 34.73 (3.65) | 29.81 (3.88) | 26.83 (4.03) |
| max shift: 1.0 LR pixels | SuperF (ours) | 39.93 (2.49) | 35.50 (2.39) | 29.49 (2.71) |
| max shift: 4.0 LR pixels | SuperF (ours) | 38.24 (2.56) | 31.49 (4.13) | 18.28 (6.37) |

## C.5 RESULTS FOR REAL SATELLITE BURSTS FROM THE WORLDSTRAT DATASET.

We demonstrated that our method can be applied to real world satellite images from the publicly available Sentinel-2 satellite images. In addition, we use 8 Sentinel-2 images included in the World-Strat (Cornebise et al., 2022) Kaggle dataset (see Fig. 9). Many of the time series included in the WorldStrat dataset are affect by noise due to lighting variation, partial cloud cover and cloud shadows, changing landcover (e.g. crops), or seasonal variations like snow cover. In these scenarios, our assumption of repeated observations of the same scene does not hold.

To investigate how to tackle these challenging scenarios, we curate two new subsets of the World-Strat dataset, namely *WorldStrat-sweet* and *WorldStrat-bitter*. The sweet dataset is composed of clean 25 time series from the WorldStrat dataset that neither have any clouds nor strong variations. In contrast, the bitter dataset, contains 25 noisy time series that show varying levels of cloud coverage and noise (see examples in Fig. 10).

To assess the effect of the real-world variability on reconstruction accuracy, we evaluate SuperF with both MSE and GNLL losses on the WorldStrat-sweet and WorldStrat-bitter subsets. As shown in Table 7, the two subsets expose clear differences, while both losses perform similarly on the clean samples in the *sweet* subset, the GNLL loss provides a consistent advantage on the noisy *bitter* samples. This confirms that modeling uncertainty is beneficial in scenarios where our assumption of repeated, noise-free observations breaks down.

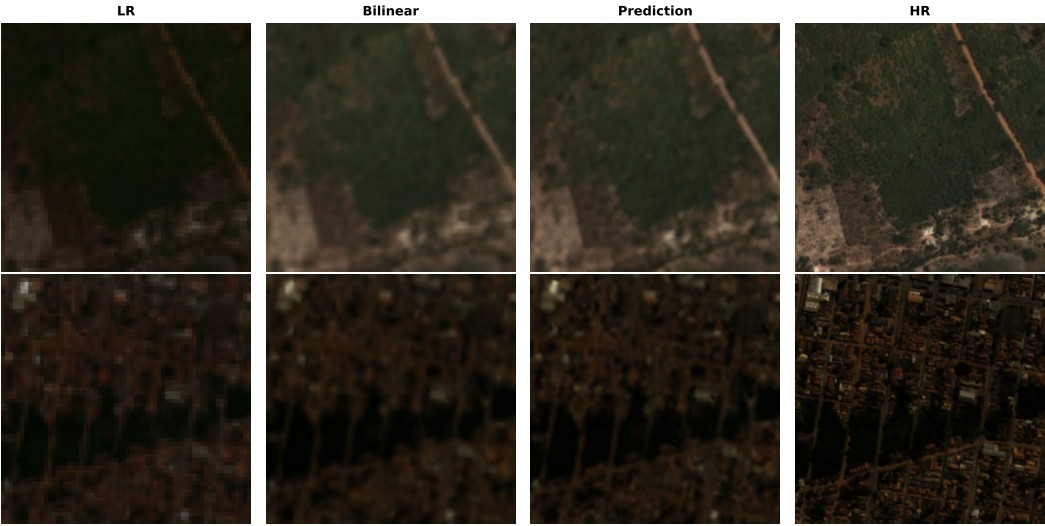

Figure 9: **Qualitative results on real WorldStrat samples.**

Table 7: **WorldStrat-sweet and WorldStrat-bitter results.** The *sweet* subset contains 25 clean time series and the *bitter* contains 25 noisy time series (e.g. cloud cover). We use the same hyper-parameters as for SatSynthBurst (see Table 4) and optimize for 2000 iterations. Standard deviation across samples is given in parentheses. For the *bitter* samples, the GNLL loss outperforms MSE. Hence, estimating the uncertainty makes SuperF more robust against noise in the image bursts (e.g. occlusions from clouds). For clean time series in *sweet*, both losses perform on par.

| Dataset | Method | PSNR | SSIM | LPIPS |
|---|---|---|---|---|
| WorldStrat-sweet | SuperF MSE | 27.86 (3.65) | 0.654 (0.151) | 0.507 (0.049) |
| | SuperF GNLL | 27.88 (3.73) | 0.657 (0.152) | 0.506 (0.051) |
| WorldStrat-bitter | SuperF MSE | 27.42 (4.54) | 0.645 (0.186) | 0.532 (0.053) |
| | SuperF GNLL | 28.46 (4.94) | 0.677 (0.182) | 0.503 (0.063) |

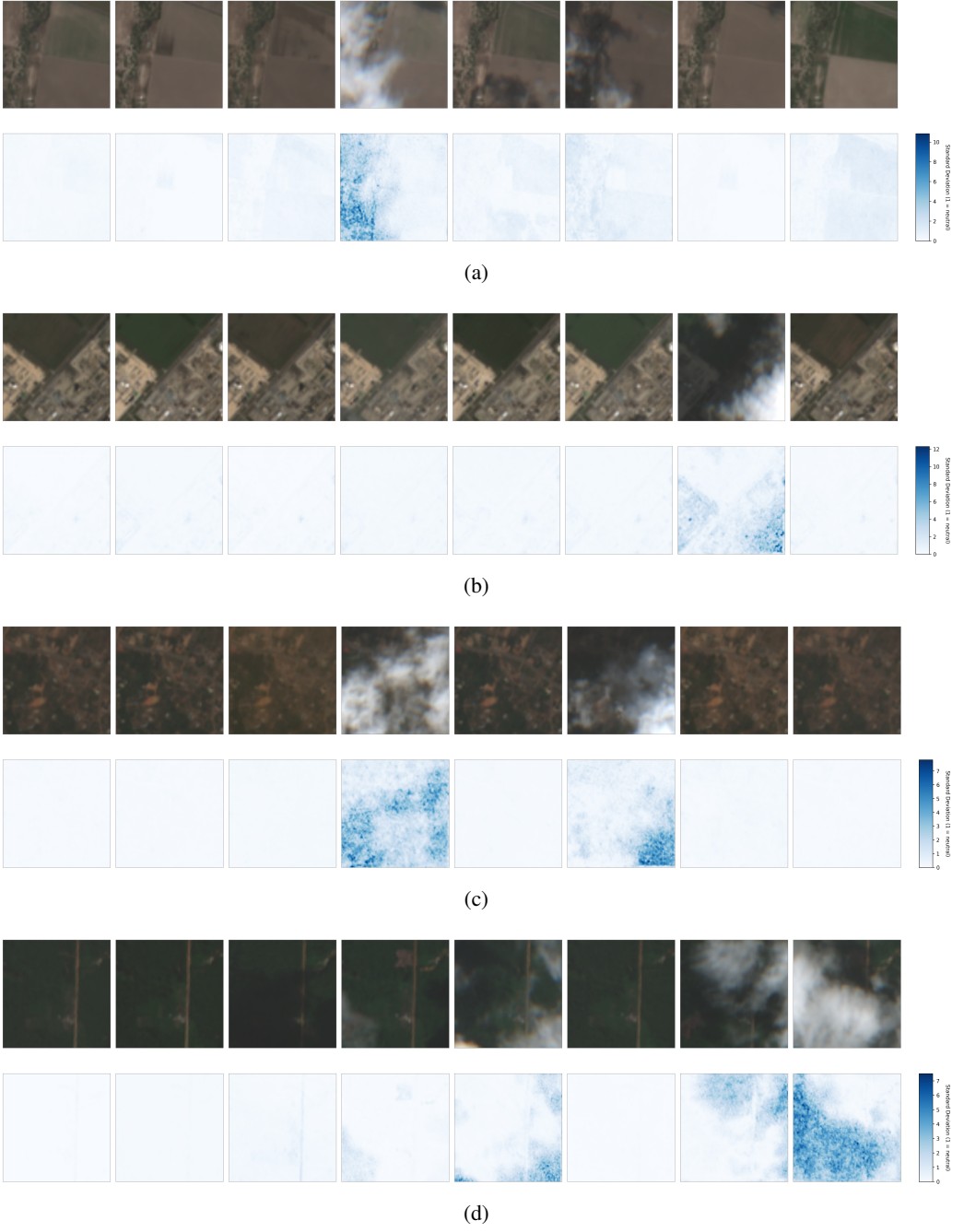

Figure 10: **Visualization of estimated uncertainty maps.** Four examples from the *WorldStrat-bitter* dataset, each showing LR frames (top row) and the corresponding estimated uncertainty maps (bottom row). The uncertainty maps highlight cloudy or inconsistent pixels, enabling the GNLL loss to downweight these during optimization.

### C.6 QUALITATIVE RESULTS FOR DIFFERENT UPSAMPLING FACTORS: $\times 2$, $\times 4$, $\times 8$

We provide additional qualitative comparisons for both satellite image and ground-level image bursts at upsampling factor $\times 2$ (Fig. 11), $\times 4$ (Fig. 12), and $\times 8$ (Fig. 13). We note that the two datasets differ in the strategy of creating versions with different upsampling factors. The SatSynthBurst dataset fixes the HR output resolution and varies the resolution of the LR frames. Hence, we expect lower performance metrics for SatSynthBurst as the upsampling factor increases. In contrast, the SyntheticBurst varies the HR output resolution with a fixed resolution of the LR frames. Thus, the metrics are not comparable across upsampling factors.

We find that for the satellite imagery, our results for the upsampling factor $\times 8$ start to become grainy. This may be a result of overfitting with a suboptimal hyperparameter setting for the Fourier feature scale.

Table 8: **Benchmarking computational costs.** Training time, speed, memory usage, and FLOPs across different upsampling factors and loss types on the SatSynthBurst dataset ($64 \times 64$ pixels low-resolution images). Values are averaged over an optimization run of 2000 iterations on a NVIDIA H100 Tensor Core GPU.

| Loss | factor | Time/Iter (ms) | Iter/s | Memory (MB) | FLOPs (G) |
|------|--------|----------------|--------|-------------|-----------|
| MSE  | $2\times$  | 2.57  | 389.49 | 186.9  | 19.40   |
| MSE  | $4\times$  | 4.39  | 227.95 | 549.2  | 77.61   |
| MSE  | $8\times$  | 11.92 | 83.92  | 1863.8 | 310.45  |
| MSE  | $16\times$ | 41.16 | 24.30  | 7250.3 | 1241.78 |
| GNLL | $2\times$  | 3.40  | 293.75 | 188.6  | 20.01   |
| GNLL | $4\times$  | 5.24  | 190.74 | 549.5  | 80.03   |
| GNLL | $8\times$  | 13.77 | 72.63  | 1864.1 | 320.11  |
| GNLL | $16\times$ | 47.38 | 21.10  | 7252.1 | 1280.44 |

### C.7 SUPER-RESOLVING ANY PLACE ON EARTH (DEMO APP)

We developed an interactive demo that allows users to select an arbitrary location on Earth and generate a super-resolved image using the SuperF technique (available on the project page: sjy-hne.github.io/superf). The demo first queries an open STAC catalog of Sentinel-2 images[9] within the user-specified time range. Each candidate image is passed through the OmniCloudMask[10] to discard observations affected by cloud cover. From the remaining cloud-free images, SuperF optimizes a single high-resolution reconstruction, using the earliest valid image as the reference frame. The user selects both the geographic location and the time interval of interest. After processing, the demo displays the resulting super-resolved image alongside all Sentinel-2 images that contributed to the reconstruction.

## D USE OF LARGE LANGUAGE MODELS

We used LLMs as search engines to support literature research and as coding assistants.

---

[9]https://earth-search.aws.element84.com/v1 (accessed: 2025-11-18)
[10]https://github.com/DPIRD-DMA/OmniCloudMask (accessed: 2025-11-18)

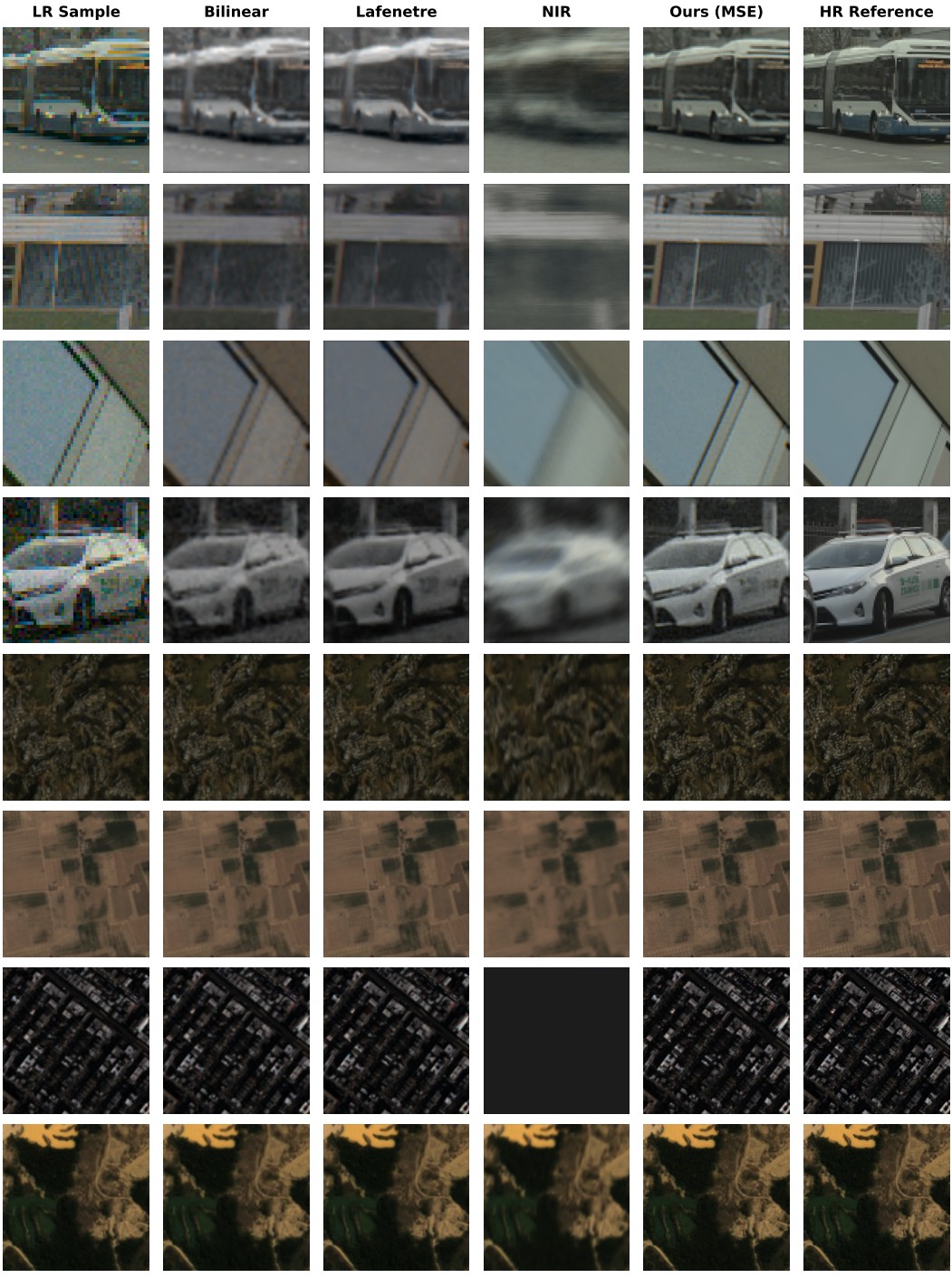

Figure 11: **Qualitative comparison with upsampling factor ×2.** From left to right, we show: one low-resolution (LR) frame, bilinear upsampling, steerable kernel regression (Lafenetre et al., 2023), NIR (Nam et al., 2022), our *SuperF* approach, and the high-resolution (HR) reference.

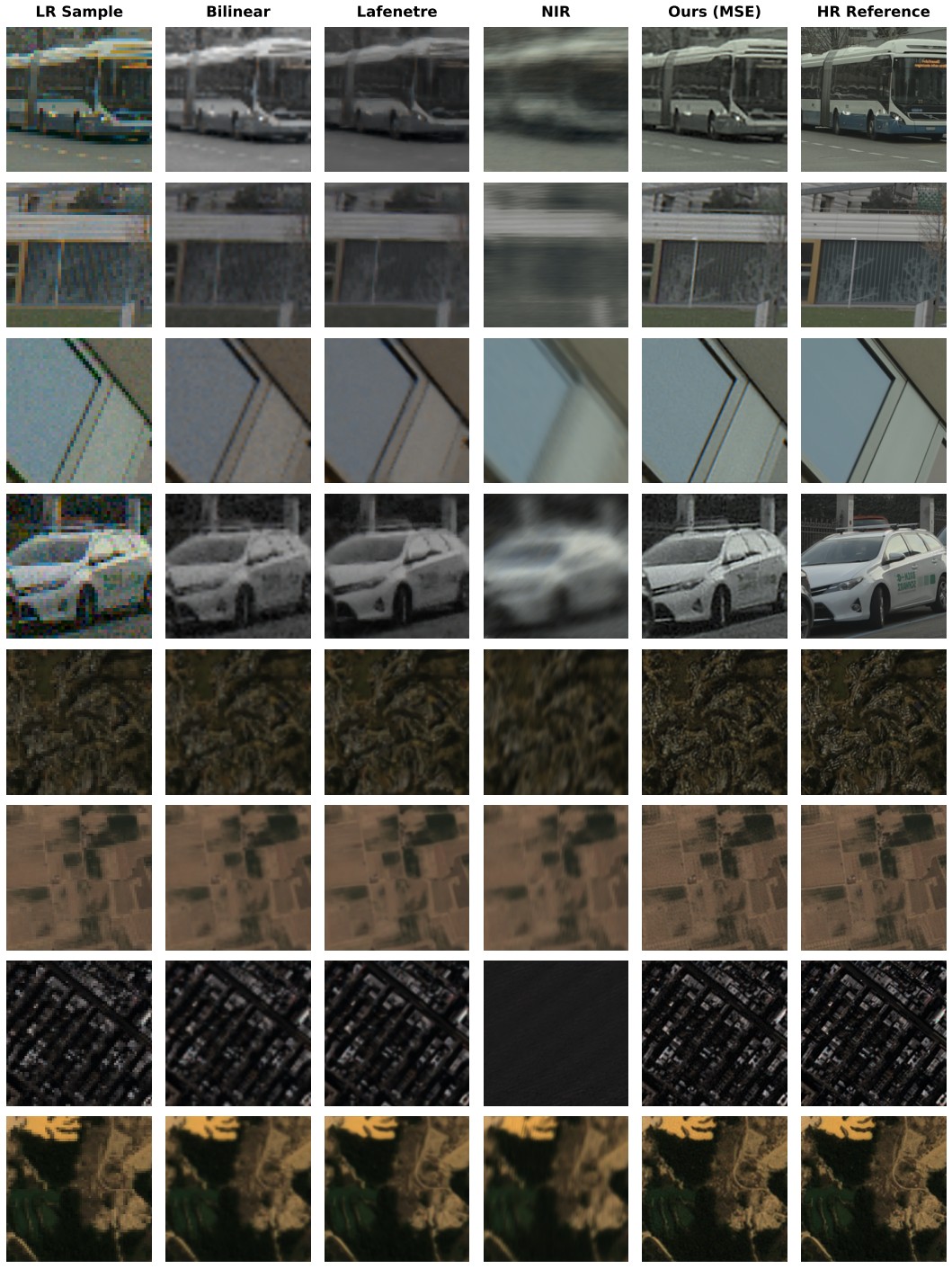

Figure 12: **Qualitative comparison with upsampling factor ×4.** From left to right, we show: one low-resolution (LR) frame, bilinear upsampling, steerable kernel regression (Lafenetre et al., 2023), NIR (Nam et al., 2022), our *SuperF* approach, and the high-resolution (HR) reference.

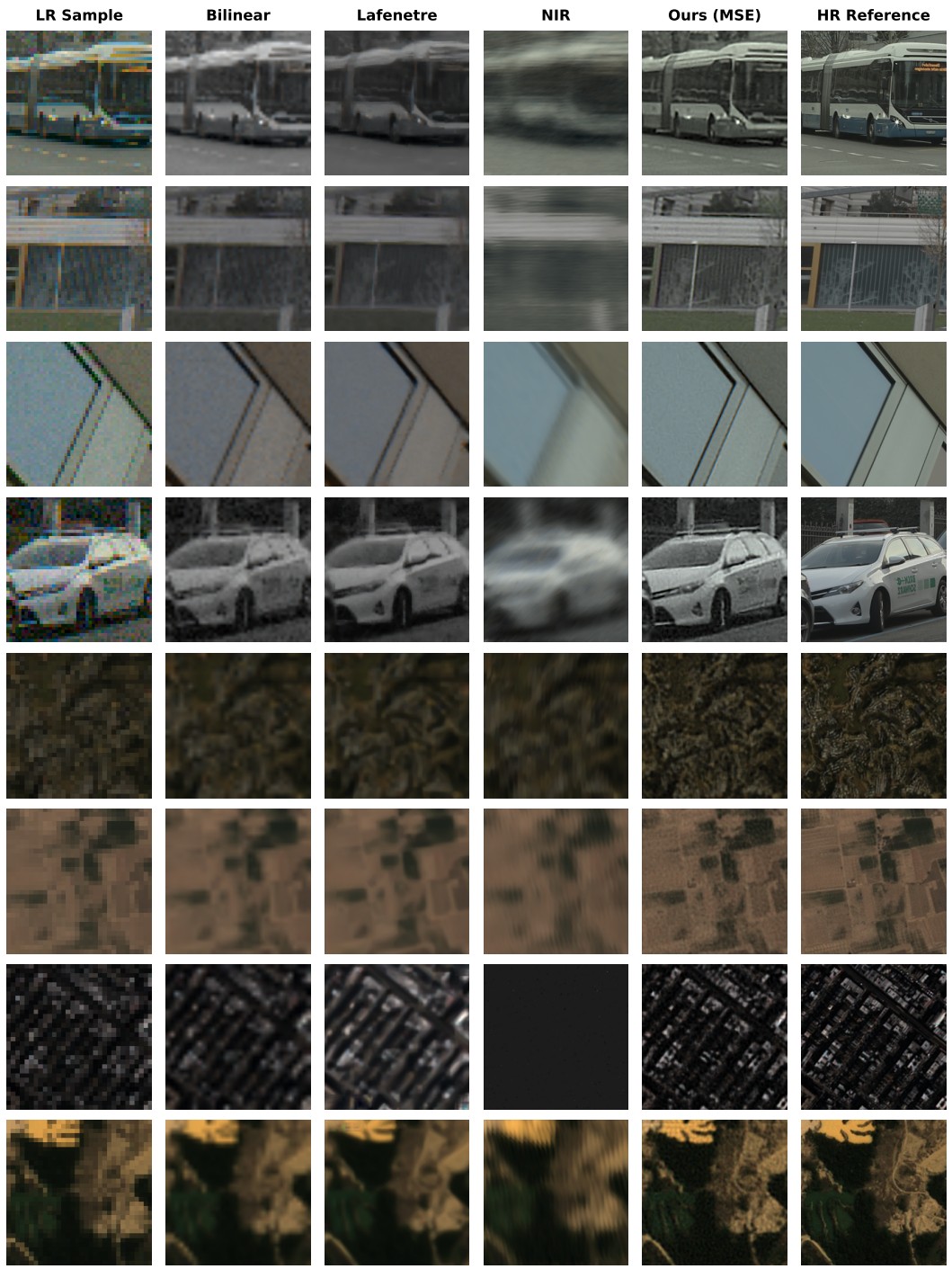

Figure 13: **Qualitative comparison with upsampling factor ×8.** From left to right, we show: one low-resolution (LR) frame, bilinear upsampling, steerable kernel regression (Lafenetre et al., 2023), NIR (Nam et al., 2022), our *SuperF* approach, and the high-resolution (HR) reference.

