# OpenReview forum: "SuperF: Neural Implicit Fields for Multi-Image Super-Resolution"
_ICLR.cc/2026/Conference — ICLR 2026 Poster_

### Official Review · Reviewer_Cozv · 2025-10-29

**Soundness:** 3
**Presentation:** 2
**Contribution:** 3
**Rating:** 4
**Confidence:** 3

**Summary:**

This paper proposed a SuperF, a test-time optimization method for multi-image super-resolution based on implicit neural representations. SuperF designs an improved sub-pixel alignment and continuous representation of the high-resolution signal based on INR. The experimental results demonstrated that proposed method achieve star-of-the-art performance.

**Strengths:**

A lot of experiments demonstrate the effectiveness of the proposed method and its components. An INR-based method is proposed for multi-image super-resolution.

**Weaknesses:**

Some detailed descriptions and reasons for the proposed method should be provided.

More experimental results should be provided for comprehensive comparison.

The writing of the paper needs improvement.

**Questions:**

Which dataset’s results are shown in Figure 2. The detailed description should be provided.

Why choose SatSynthBurst and SyntheticBurst datasets for evaluation? Why build SatSynthBurst from the WorldStrat dataset? Can the WorldStrat dataset be used as an evaluation dataset? Did the previous method also use the same configuration?

In Table 1 and Table 2, what does the value in parentheses mean? The detailed description should be presented in the caption of the Table. The complexity of the proposed method should be provided.

Why the proposed (MSE) and the proposed (GNLL) have different performance. GNLL-based model has the best results in most cases for the SatSynthBurst dataset. However, MSE-based model has the best results for the SatSynthBurst dataset and large margins. In Figure 5, this phenomenon also occurs. The reason should be provided. What do these results represent?

Why use two loss functions, i.e., mean squared error(MSE) and the Gaussian negative log-likelihood (GNLL)? Do previous methods also use two loss functions?

The author claims that SuperF is a test-time optimization method. How to demonstrate the test-time optimization for the proposed method? Relevant experiments and analysis should be provided

Why use 4 MLP layers in the INR decoder and 3 MLP layers in the Uncertainty decoder? Corresponding ablation studies should be provided for specific layers.

The compared methods, i.e., Nametal.(2022), Lafenetreetal.(2023), they were two years ago, the latest methods should be discussed and compared

Some typos should be revised, such as the left quotation mark in line 018. Please review the entire manuscript.

---

> ### Author Response · Authors · 2025-11-24
>
> We thank the reviewer for the time and feedback. We respond to each question below. Please let us know if these clarifications and revisions have resolved your concerns. The revised pdf contains the updates highlighted in dark-blue.
> Since this review gave the lowest score compared to the other three reviews, we would like to resolve your questions to improve our manuscript. Thanks for your feedback.
>
> **Figure 2**
>
> Thanks for pointing this out. The results in Figure 2 show samples from the SyntheticBurst (row 1, 2) and SatSynthBurst (row 3, 4). We clarified this in the caption of Figure 2 in the revised pdf.
>
> **Datasets**
>
> While prior work on MISR typically focuses only on one domain, our goal was to demonstrate that our proposed method can generalize to different domains. Previous methods used private datasets that are not published. Therefore, we run the baseline methods on the selected datasets ourselves.
>
> SyntheticBurst is a ground-level dataset used in the MISR literature, originally introduced for supervised learning. We use this dataset to study test-time optimization that does not assume access to any high-resolution training data.
> Our presented SatSynthBurst dataset fills a gap, as there is no publicly available synthetic MISR dataset with satellite images. We discussed already in line 295 (revised pdf) that SatSynthBurst allows us to control the sub-pixel shifts and to create low-resolution base frames that perfectly align with the high-resolution test image for clean evaluation. Creating MISR datasets by pairing real data is challenging, as there is a spectral and spatial misalignment between low-resolution base frames and the high-resolution reference images. This misalignment also exists e.g. in WorldStrat.
>
> However, we added additional experiments. We make use of the real WorldStrat data and create two curated subsets WorldStrat-sweet (25 clean time series) and WorldStrat-bitter (25 noisy time series) to study the robustness against occlusions from partial cloud cover. See section 4.5 (line 473-478) and section C.5.
>
>
> **Table 1 and Table 2**
>
> As written at the end of the caption of Table 1 and 2: “Standard deviation across samples shown in parentheses.” (see submission pdf)
>
> **MSE vs. GNLL**
>
> We agree that a better discussion was needed. We discuss these results in line 369-372:
>
> *“While on SyntheticBurst both losses perform on par, the GNLL loss consistently improves upon the MSE on SatSynthBurst. The GNLL possibly allows to be more robust against the spectral variation in this dataset, which is less pronounced in the ground-level bursts.”*
>
> We also discuss the additional experiments shown in section C.5, Table 7 in line 473-479:
>
> *“**Robustness to occlusions**. To study robustness of our approach for noisy satellite image time series where some LR frames are partially covered by clouds, we curated two subsets of the WorldStrat dataset (Cornebise et al., 2022) using real Sentinel-2 images, WorldStrat-sweet and WorldStrat- bitter, with clean and noisy time series, respectively. On the bitter dataset, we find that the GNLL approach effectively ignores cloudy pixels (see Appendix Figure 10) and improves performance compared to using the MSE loss (see Appendix Table 7). For clean time series in sweet, both losses perform on par.”*
>
> **Loss functions**
>
> We apologize if that was not clear. We use only one loss function at a time. We use either MSE or GNLL. We clarified this in section 3.3 (revised pdf).
>
> **TTO**
>
> This question is not fully clear to us. The here proposed method does not require any high-resolution training images. We only optimize the INR model on the given low-resolution frames at test time. This characteristic allows us for example to apply our method to real satellite data to super-resolve any place on Earth (see section C.7).
>
> **Uncertainty decoder**
>
> We have simplified and improved our uncertainty estimation approach, which does not require any additional decoder anymore. We simply output the uncertainty maps from the main INR decoder. We updated the revised pdf section 3.3 and included the uncertainty estimation in the architecture diagram shown in Figure 1.
>
> Our updated approach follows a pure INR paradigm and uses the coordinate-based neural network itself to represent additional frame-specific uncertainty maps as visualized in Figure 1. Examples of how these uncertainty maps capture occlusions by clouds in real satellite time series are shown in the Appendix Figure 10. This new approach is simpler and more stable than the initial approach (see updated results in Table 1 “SuperF GNLL”). Hence, we decided to drop the results from the initial approach.
>
> **Baselines**
>
> We compare against the most related and latest baselines we could find in the literature. We appreciate any guidance on existing TTO MISR baselines to compare to.
>
> **Typos**
>
> Thanks for pointing us to the typos.

---

### Official Review · Reviewer_gFfE · 2025-10-30

**Soundness:** 3
**Presentation:** 3
**Contribution:** 3
**Rating:** 6
**Confidence:** 3

**Summary:**

This paper presents SuperF, a test-time optimization method for multi-image super-resolution (MISR) using implicit neural representations (INRs). It improves sub-pixel alignment of low-resolution frames without requiring high-resolution training data, avoiding "hallucinated" structures. SuperF outperforms existing methods on both satellite and ground-level images, offering a data-efficient solution for super-resolution.

**Strengths:**

The use of implicit neural representations (INRs) for multi-image super-resolution (MISR) is innovative, offering a novel approach that leverages sub-pixel alignment optimization.
By avoiding the need for high-resolution training data, SuperF is a practical solution for real-world applications, including satellite imagery and handheld cameras.
The approach generalizes well across different datasets without the need for retraining, demonstrating versatility in various contexts like remote sensing and environmental monitoring.

**Weaknesses:**

The iterative optimization process, although memory-efficient, can be time-consuming, which might limit its applicability in real-time or mobile scenarios. Moreover, the method's performance is somewhat dependent on the Fourier feature scale hyperparameter, which requires careful tuning for different domains.

**Questions:**

1.	The current approach assumes that all frames depict the same scene, but how does SuperF perform when there are significant changes between frames, such as moving objects or occlusions?
2.	While the method shows domain sensitivity, could there be an automatic or adaptive mechanism for adjusting the scale based on the input domain, thus reducing the need for domain-specific tuning?
3.	Given that cloud cover can obscure parts of satellite images, how well does SuperF perform when parts of the scene are occluded or affected by noise, especially with respect to the uncertainty estimation?

---

> ### Author Response · Authors · 2025-11-24
>
> We thank the reviewer for the time and the feedback. We reply to the individual points below. The revised pdf contains the updates highlighted in dark-blue. Please let us know if these clarifications and updates resolve your questions.
>
> **Robustness to occlusions**
>
> That is correct, we assume repeated observations of static scenes. We provide additional experiments for two curated subsets of the real WorldStrat data in section C.5 in the appendix (revised pdf).
>
> - _WorldStrat-sweet_ contains 25 clean satellite image time series.
>
> - _WorldStrat-bitter_ contains 25 noisy satellite image time series with partial cloud cover.
>
> We find that the GNLL loss with the uncertainty estimation allows our approach to be more robust against high noise levels (e.g. occlusions from clouds) and leads to better performance on _WorldStrat-bitter_ (see section 4.5, line 473-478).
>
> **Automatic tuning of the Fourier Feature scale**\
> We agree that an auto-tuning approach would facilitate the transfer to new domains. We tried optimizing the FF scale parameter directly together with the INR model. However, even when initialized with an optimal value, the model reduces the FF scale which leads to overly smooth outputs. The problem is that when optimizing SuperF, there is no high-resolution supervision that could inform the model about an optimal parameter. The only training signal comes from multiple shifted LR images.
>
> However, as our results show this hyperparameter is quite robust within a domain, e.g. satellite images or ground-level images. We added the following sentence to the discussion of limitations (line 518):
>
> _“In practice, this parameter can be tuned by using some high-resolution validation images as shown in Figure 7 or via visual inspection (see Figure 8).”_
>
> **Role of uncertainty estimation for occlusions**
>
> We visualize the estimated uncertainty maps per LR frame and show that they can represent noise coming from occlusions such as cloud cover in satellite images in Figure 10 in the appendix (revised pdf).  We discuss these results in section 4.5 line 473-479.
> This can explain the performance difference between WorldStrat-sweet vs. WorldStrat-bitter, as the model effectively ignores noisy pixels that are occluded by clouds.

---

> > ### Comment · Reviewer_gFfE · 2025-11-26
> >
> > Thanks for your response
> >
> > Overall, the responses are clear and thorough. The additional experiments, uncertainty visualizations, and discussion on the Fourier Feature scale satisfactorily address my main concerns. Therefore, I would like to maintain my score and lean toward accepting this paper.

---

> > > ### Author Response · Authors · 2025-11-28
> > >
> > > Thank you for the response. We are glad that the reviewer finds our responses and revision helpful to resolve their concerns. It is encouraging to read that they consider our proposed approach to be "innovative" and a "practical solution for real-world applications".

---

### Official Review · Reviewer_hY6U · 2025-11-01

**Soundness:** 3
**Presentation:** 3
**Contribution:** 3
**Rating:** 6
**Confidence:** 4

**Summary:**

The paper introduces SuperF, a test-time optimization framework for multi-image super-resolution (MISR) using implicit neural representations (INRs). It jointly optimizes a shared coordinate-based MLP and frame-specific affine alignments to reconstruct a high-resolution image from multiple low-resolution inputs. The approach requires no training data and is validated on both synthetic satellite and handheld burst datasets, demonstrating strong PSNR and SSIM improvements over prior TTO baselines.

**Strengths:**

SuperF is well-motivated, bridging the gap between INR-based image representation and multi-frame super-resolution. The joint optimization of alignment and representation is elegant and mathematically consistent. The experiments are thorough, covering domain generalization, ablations, and sensitivity analyses. The work also provides a new synthetic satellite burst dataset, adding clear community value.

**Weaknesses:**

1. The paper lacks direct visual or quantitative comparison with modern learning-based MISR networks (e.g., DeepBurstSR or Transformer variants), limiting practical benchmarking.
2. The uncertainty decoder’s effect is minor and under-analyzed; more discussion or visualization would clarify its contribution.
3. Runtime (several seconds per image) and computational cost are only briefly mentioned; profiling across resolutions would make claims about efficiency more credible.
4. The sensitivity to Fourier feature scale is described, but a cross-domain auto-tuning or normalization strategy would strengthen robustness.
5. The method assumes static scenes; discussion on handling dynamic or partially misaligned inputs (e.g., moving clouds, handheld motion) is needed for real-world applicability.

**Questions:**

See weaknesses for details.

---

> ### Author Response · Authors · 2025-11-24
>
> Thank you for your time and the constructive feedback. We clarify individual points below. The revised pdf contains the updates highlighted in dark-blue. Please let us know if these clarifications and updates resolve your questions.
>
> **Effect of uncertainty estimation:**
>
> We agree that further analysis of the uncertainty estimation was needed.
>
> The revised pdf contains an extended analysis and discussion of the uncertainty estimation in section 3.3 and section 4.5 (line 473-478).
>
> We have simplified and improved our uncertainty estimation approach, which does not require any additional decoder anymore. We simply output the uncertainty maps from the main INR decoder.
>
> We provided additional experiments for two curated subsets of the real WorldStrat data in section C.5 in the appendix (revised pdf).
>
> - _WorldStrat-sweet_ contains 25 clean satellite image time series.
>
> - _WorldStrat-bitter_ contains 25 noisy satellite image time series with partial cloud cover.
>
> We find that the GNLL loss with the uncertainty estimation allows it to be more robust against high noise levels and leads to better performance on _WorldStrat-bitter_.
>
> Furthermore, we visualized estimated uncertainty maps on these real satellite time series and show that they can represent noise coming from occlusions such as cloud cover in satellite images in Figure 10 (revised pdf). This allows to ignore noisy pixels during the optimization.
>
> **Runtime across resolutions:**
>
> Thanks for this suggestion. We provide the analysis of runtime across upsampling factors in Table 8 in the revised pdf and already discussed it in the limitation section (see line 507).
>
> **Sensitivity of Fourier Feature (FF) Scale:**
>
> We agree that an auto-tuning approach would strengthen robustness. We tried optimizing the FF scale parameter directly together with the INR model. However, even when initialized with an optimal value, the model reduces the FF scale which leads to overly smooth outputs. The problem is that when optimizing SuperF, there is no high-resolution supervision that could inform the model about an optimal parameter. The only training signal comes from multiple shifted LR images.
>
> However, as our results show this hyperparameter is quite robust within a domain, e.g. satellite images or ground-level images. We added the following sentence to the discussion of limitations (line 518):
>
> _“In practice, this parameter can be tuned by using some high-resolution validation images as shown in Figure 7 or via visual inspection (see Figure 8).”_
>
> **Comparison with supervised baselines:** \
> This work focuses on unsupervised test-time optimization approaches that do not rely on high-resolution training data. Supervised approaches that rely on high-resolution training data are not applicable under our studied problem setup, which assumes that there is no available high-resolution training data (as stated in line 80, line 113).
>
> **Real world applicability:**
>
> Exactly, we assume repeated observations of static scenes. Our method utilizes sub-pixel shifts that can originate from either handheld motion (ground-level imagery) or from imperfect georeferencing (satellite imagery). Hence, handheld motion is considered a required characteristic for our method and not seen as noise.
>
> In contrast, partial cloud cover introduces additional noise, which we analyze in new experiments by comparing _WorldStrat-sweet_ and _WorldStrat-bitter_ (in section C.5 see above)_._
>
> To further test SuperF’s real world applicability, we developed an interactive demo app to super-resolve any place on Earth (see section C.7 and Figure 14 of the revised pdf for screenshots of the tool). This demo simply takes any geographic coordinate as the input, downloads multiple Sentinel-2 satellite images (available every 5 days), filters out cloudy images, and then runs SuperF to compute a super-resolved image. Qualitative examples with upsampling factor 5 are shown in Figure 3 in the revised pdf. This demo will be made publicly available upon publication along with the developed code and datasets.

---

### Official Review · Reviewer_qyd9 · 2025-11-10

**Soundness:** 3
**Presentation:** 3
**Contribution:** 3
**Rating:** 6
**Confidence:** 4

**Summary:**

This paper presents SuperF, a new method for improving multiple-input super resolution (MISR).
SuperF uses coordinate-based neural networks that represents images as continuous signals rather than fixed pixels.
It performs test-time optimization and  does not require pre-trained high-resolution data.
SuperF shares one neural representation across  T low-resolution (LR) frames and jointly optimizes both alignments and reconstruction.

SuperF sbegins with T LR frames, each frame  defined on a finite set of points.
The method estimates the underlying signal at a denser set of points relative to  LR input frames.
It assumes that each LR  frame is a convolution of a boxcar filter withan affine transformation of the high-resolution (HR)  target.
To merge multiple frames, the method aligns the LR frames at the sub-pixel level.

SuperF achieve its goal by aligning the the LR frames using affine transformation while modeling the HR image as a continuous signal through the neural network.
The HR prediction is blurred with a boxcar filter and downsampled to match the LR resolution.
The optimization for SuperF focuses on adjusting both the neural network parameters and the affine transformations so that all the LR frames agree on the same HR signal.

This paper provides experimental result to support it claims.

**Strengths:**

The paper is well organized and clearly presented. It provide sufficient information to understand the motivation of the work. It also provides adequate information about prior and related work. This help in establishing the originality of the work. The approach of this work as an test-time optimization also underscores the significance of the contribution. The method provides a solution to the need for resources-efficient methods deployment of machine learning methods, especially on resource-constrained hardware. Finally, the experiments are well designed, clearly presents, and supports the central claims of the paper. This reflects the overall quality of the paper.

**Weaknesses:**

1).  Algorithm Statement or Architecture Diagram: While this paper did a good job at explaining the components of SuperF, it does not provide enough information to understand how these components fit together, especially for the purpose of reproducibility. For example, Table 5 in the Appendix shows that there are two decoder networks (Fourier feature-based decoder and  uncertainty decoder). It is not clear how they come together as a singe model under the SuperF model architecture. Having an algorithm statement or a complete model architecture will improve the reproducibility and the potential impact of SuperF.

2). MSE vs GNLL:  Lines 245-246 state that  you have used the GNLL instead of  mean squared error. What was the assumption on the parameters of the  underlying Gaussian distribution for the GNLL?

**Questions:**

Please check my comments under the Weaknesses.

---

> ### Author Response · Authors · 2025-11-24
>
> Thank you for your time and the constructive feedback. We respond to the two comments individually below. The revised pdf contains the updates highlighted in dark-blue. Please let us know if these clarifications and updates resolve your questions.
>
> **1) Algorithm Statement or Architecture Diagram:**
>
> We appreciate the suggestions to improve the presentation of the overall methodology. We agree that clarification was needed on how we integrate the uncertainty estimation for the GNLL loss. We updated the revised pdf section 3.3 and included the uncertainty estimation in the architecture diagram shown in Figure 1.
>
> To summarize, we have simplified and improved our uncertainty estimation approach, which does not require any additional decoder anymore. We simply output the uncertainty maps from the main INR decoder.
>
> Our _updated approach_ follows a pure INR paradigm and uses the coordinate-based neural network itself to represent additional frame-specific uncertainty maps as visualized in Figure 1. Examples of how these uncertainty maps capture occlusions by clouds in real satellite time series are shown in the Appendix Figure 10. This new approach is simpler and more stable than the initial approach. Hence, we decided to drop the results from the initial approach.
>
> For completeness, _our initial approach_ was based on a separate uncertainty decoder _h,_ which is another ReLU MLP with 3 layers. This decoder takes as input the pixel-wise RGB reflectance from the estimated LR images and the target LR frame.
>
> For reproducibility, we release all code and data upon publication.
>
> **2) Assumptions for MSE and GNLL loss**
>
> Optimizing the Gaussian Negative Log Likelihood (GNLL) can be seen as a generalization of optimizing the mean squared error (MSE). Formally, optimizing the MSE is equivalent to parameterizing the conditional distribution as a Gaussian normal distribution with a fixed noise term (Nix and Weigend, 1994). Hence, optimizing the MSE is a special case of optimizing the GNLL and equivalent if the variance is constant (called homoscedastic uncertainty).
>
> Hence, the derivations of the loss functions make the assumptions that the output has Gaussian noise with i) constant variance for MSE and ii) varying noise for GNLL (called heteroscedastic uncertainty). That is, the assumptions when deriving MSE are stronger than when deriving GNLL. We clarified this assumption in line 259 of the revised pdf.
>
> We added a reference to Nix and Weigend (1994), who brought forward optimizing GNLL in the neural network community.
>
> **References**
>
> Goodfellow, Ian, et al. _Deep learning_. Vol. 1. No. 2. Cambridge: MIT press, 2016.
>
> David A Nix and Andreas S Weigend. Estimating the mean and variance of the target probability distribution. In ICNN, 1994.

---

### Author Response · Authors · 2025-11-24
**Summary of updates**

Dear AC, dear reviewers,

We thank you for the constructive feedback. We responded in detail to all questions below and uploaded a revised pdf (highlighting updates in dark blue). All references in the rebuttal (e.g. line numbers) refer to the new revised pdf. We are happy to answer further questions during the discussion phase.

**Summary of the major updates:**
1. Revised the methodology section and Figure 1 to better explain the uncertainty estimation.
2. Additional experiments on real satellite image time series with new datasets to show robustness.
3. Explained how the Fourier feature scale hyperparameter can be chosen in practice.

---

### Author Response · Authors · 2025-12-02
**Final Comment to the AC**

Dear AC,

We thank all reviewers again for their feedback that helped us to improve the manuscript.
Here we summarize the strengths and weaknesses highlighted by the reviewers and our clarifications incorporated in the revised pdf.

We are pleased to receive positive feedback highlighting that our approach is “_elegant_” (reviewer hY6U), “_innovative_” (reviewer gFfE), and “_effective_” (reviewer Cozv), and that the generalization capabilities of our test-time optimization approach that does not require any training on high resolution data is perceived as a “_practical solution for real-world applications_” (reviewer gFfE).
It is motivating to see consensus that our manuscript is “_well-motivated_” (reviewers hY6U, qyd9) and “_well organized and clearly presented_” (reviewer qyd9). Furthermore, it is noted that our experiments “_are well designed_” (reviewer qyd9) and “_thorough_” (reviewers hY6U, Cozv). Lastly, our provided dataset adds “_clear community value_” (reviewer hY6U).

Major concerns were addressed by revising the methodology section to clarify the estimation of uncertainty and by adding additional experiments and analyses to discuss the robustness for real satellite imagery (reviewers qyd9, hY6U, gFfE).
In addition, we have clarified how the Fourier feature scale hyperparameter can be chosen in practice (reviewers hY6U, gFfE).

In the only rebuttal comment we received before the reported leak, reviewer gFfE stated that our changes "_satisfactorily address my \[the reviewer’s] main concerns_“.

We thank the AC for their important work and consideration of our work and rebuttal.

---

### Meta-Review · Area_Chair_YnPo · 2026-01-02

**Summary:**

This paper proposes a test-time optimization method for MISR based on implicit neural representations. The major concerns of this paper include the unclear architecture diagram, lack of direct visual comparisons with recent learning-based MISR networks, limited analysis of the uncertainty decoder’s effect, the rationality of the assumption, and robustness of the proposed method.

**Reviewer Concerns:**

The major concerns of this paper including the unclear architecture diagram, lack of direct visual comparisons with recent learning-based MISR networks, limited analysis of the uncertainty decoder’s effect, the rationality of the assumption, and robustness of the proposed method should be addressed in the rebuttal.

**Reviewer Scores:**

In the rebuttal and provided revised paper, the authors solve the above-mentioned problems.

---

### Decision · Program_Chairs · 2026-01-26

Accept (Poster)